# PTRN-1, a microtubule minus end-binding CAMSAP homolog, promotes microtubule function in *Caenorhabditis elegans* neurons

**Claire E Richardson[1], Kerri A Spilker[1†], Juan G Cueva[2], John Perrino[3], Miriam B Goodman[2], Kang Shen[1,4]***

[1]Department of Biology, Stanford University, Stanford, United States; [2]Department of Molecular and Cellular Physiology, Stanford University, Stanford, United States; [3]Cell Sciences Imaging Facility, Stanford University, Stanford, United States; [4]Howard Hughes Medical Institute, Stanford University, Stanford, United States

**Abstract** In neuronal processes, microtubules (MTs) provide structural support and serve as tracks for molecular motors. While it is known that neuronal MTs are more stable than MTs in non-neuronal cells, the molecular mechanisms underlying this stability are not fully understood. In this study, we used live fluorescence microscopy to show that the *C. elegans* CAMSAP protein PTRN-1 localizes to puncta along neuronal processes, stabilizes MT foci, and promotes MT polymerization in neurites. Electron microscopy revealed that *ptrn-1* null mutants have fewer MTs and abnormal MT organization in the PLM neuron. Animals grown with a MT depolymerizing drug caused synthetic defects in neurite branching in the absence of *ptrn-1* function, indicating that PTRN-1 promotes MT stability. Further, *ptrn-1* null mutants exhibited aberrant neurite morphology and synaptic vesicle localization that is partially dependent on *dlk-1*. Our results suggest that PTRN-1 represents an important mechanism for promoting MT stability in neurons.

**\*For correspondence:**
kangshen@stanford.edu

**†Present address:** Metabolic Engineering and Systems Biology, Biogen Indec, Cambridge, United States

**Competing interests:** The authors declare that no competing interests exist.

**Reviewing editor**: Oliver Hobert, Columbia University, United States

## Introduction

In neurons, microtubules (MTs) provide structural support, provide tracks that molecular motors use to transport cargo from the cell body to the synapses, and promote the establishment and maintenance of neuronal polarity. The MT bundles in neuronal processes, especially axons, are exceptionally stable compared to those present in most cell types (*Conde and Cáceres, 2009*). Many proteins bind along the side or at the plus end of neuronal MTs to promote MT stability (*Conde and Cáceres, 2009*). Additionally, tubulin posttranslational modifications contribute to the structure and function of neuronal MTs (*Janke and Kneussel, 2010*). A long-standing question is what mechanisms prevent depolymerization from the MT minus ends.

MTs are polarized, cylindrical structures assembled from α/β-tubulin heterodimers. Although tubulin dimers can be added and removed from the plus end of an MT, the minus end depolymerizes continuously if not stabilized (*Mimori-Kiyosue, 2011*). In most cells, minus ends are anchored at the centrosome by the γ-tubulin ring complex (γ-TuRC). Ninein, another minus end-binding protein, both stabilizes MTs that have been released from the centrosome and anchors MTs at centrosomal and non-centrosomal sites (*Mogensen et al., 2000*).

To produce the MT bundles in neurites, MTs are nucleated at the centrosome and transported into neurites by MT motor proteins (*Yu et al., 1993*; *Ahmad et al., 1998*; *Wang and Brown, 2002*). Recently, *Ori-McKenney et al. (2012)* showed that, in the dendritic arbor of *D. melanogaster* neurons, minus ends are also both nucleated and stabilized by γ-tubulin localized to Golgi outposts. Still, in at

**eLife digest** Microtubules are tiny tubular structures made from many copies of proteins called tubulins. Microtubules have a number of important roles inside cells: they are part of the cytoskeleton that provides structural support for the cell; they help to pull chromosomes apart during cell division; and they guide the trafficking of proteins and molecules around inside the cell. Most microtubules are relatively unstable, undergoing continuous dis-assembly and re-assembly in response to the needs of the cell. The microtubules in the branches of nerve cells are an exception, remaining relatively stable over time. Now Richardson et al. and, independently, Marcette et al., have shown that a protein called PTRN-1 has an important role in stabilizing the microtubules in the nerve cells of nematode worms.

By tagging the PTRN-1 proteins with fluorescent molecules, Richardson et al. were able to show that these proteins were present along the length of the microtubules within the nerve cells. Further work showed that the PTRN-1 proteins stabilize the microtubule filaments within the branches of these nerve cells and also hold them in position.

Richardson et al. also found that worms that had been genetically modified to prevent them from producing PTRN-1 failed to traffic certain molecules to the synapses between nerve cells. Moreover, these mutants also had problems with the branching of their nerve cells; however, these defects were relatively mild, which suggests that other molecules and proteins act in parallel with PTRN-1 to stabilize microtubules in nerve cells. Further work should be able to identify these factors and elucidate how they work together to stabilize the microtubules in nerve cells.

least some cell types, γ-tubulin could not be detected in neurites (*Baas and Joshi, 1992*). Further, the centrosome is dispensable for promoting neuronal MT function in both *D. melanogaster* (*Basto et al., 2006*) and cultured hippocampal neurons (*Stiess et al., 2010*). These studies imply that additional factors stabilize the minus ends of MTs released from the centrosome and nucleate MTs in neurites.

The CAMSAP family of proteins has been identified as a group of conserved, MT minus end-binding proteins (*Baines et al., 2009*). Patronin, the CAMSAP homolog in *D. melanogaster*, promotes MT stability by protecting minus ends released from the centrosome from depolymerization by kinesin-13 MT depolymerase (*Goodwin and Vale, 2010*; *Wang et al., 2013*). In *H. sapiens* epithelial cells, CAMSAP3 (Nezha) stabilizes MT minus ends at adherens junctions and throughout the cytosol (*Meng et al., 2008*). Along with the partially redundant CAMSAP2, CAMSAP3 promotes proper MT organization and organelle assembly (*Tanaka et al., 2012*). Importantly, both Patronin and CAMSAP3 have been shown to bind the MT minus end directly in vitro (*Meng et al., 2008*; *Goodwin and Vale, 2010*). Meng et al. purified and fluorescence-labeled the C-terminal half of CAMSAP3 and sequentially added rhodamin-labeled and rhodamin-unlabeled MTs (*Meng et al., 2008*). The CAMSAP3 fragment colocalized with the end of the MT with higher rhodamine fluorescence, which indicates that it was bound to the minus end (*Meng et al., 2008*). Goodwin and Vale found that purified GFP–Patronin, which was attached to a coverslip bound and anchored rhodamine-MTs by a single end (*Goodwin and Vale, 2010*). Further, they used MT gliding assays in which either the plus-end motor kinesin or the minus-end motor dynein were added to the purified rhodamine-MT plus GFP–Patronin to show that the Patronin-bound end of the MT was the minus end (*Goodwin and Vale, 2010*). Taken together, this literature suggests that the CAMSAP family of proteins plays important roles in stabilizing MTs in vivo.

We tested the hypothesis that CAMSAP proteins bind and stabilize MT minus ends in neuronal processes. We used *C. elegans* because neurite structure and function, along with subcellular protein localization, can be readily observed in vivo. Live imaging of the *C. elegans* CAMSAP homolog, PTRN-1, in cells co-labeled with fluorescence-tagged MTs indicates that PTRN-1 localizes to MT-binding puncta throughout neuronal processes. Using a combination of live imaging and electron microscopy, we implicate a role for PTRN-1 in promoting MT stability and polymerization in neurites. Finally, we show that the loss of *ptrn-1* function results in defective neurite branching and mislocalization of synaptic vesicles, indicating that it has an important role in neuron morphology and function. The loss of the DLK-1 pathway, which is known to function in synapse localization and neurite morphology (*Nakata et al., 2005*; *Tedeschi and Bradke, 2013*), partially suppresses these defects.

# Results

## PTRN-1 exhibits punctate localization throughout neuronal processes

The *C. elegans* genome encodes a single homolog of the CAMSAP family of MT minus end-binding proteins, PTRN-1. (*Figure 1—figure supplement 1A*). PTRN-1a has a conserved domain structure with previously characterized CAMSAP proteins *H. sapiens* CAMSAP3 and *D. melanogaster* Patronin, consisting of a calponin homology domain near the N-terminus, a central region with predicted coiled-coil repeats, and a C-terminal CKK domain (*Figure 1—figure supplement 1B*) (*Meng et al., 2008*; *Goodwin and Vale, 2010*). As the other PTRN-1 isoform, PTRN-1b, lacks the CKK domain, which is the domain required for MT binding in other CAMSAP proteins, we focused on the PTRN-1a isoform (*Meng et al., 2008*; *Baines et al., 2009*; *Goodwin and Vale, 2010*). Using a fosmid expressing mCherry from the *ptrn-1* promoter (*Tursun et al., 2009*), we observed *ptrn-1* expression in many tissues throughout development, including neurons (*Figure 1—figure supplement 1C–H*).

We examined PTRN-1a subcellular localization in neurons using fluorescence-tagged PTRN-1a. Three fluorescence-tagged PTRN-1 constructs - PTRN-1a::YFP and PTRN-1a::tdTomato, which both used C-terminal tags, and GFP::PTRN-1, in which PTRN-1 was tagged at the N-terminus – localized to small, closely spaced puncta throughout neurites (*Figure 1—figure supplement 2*, *Figure 1*). We focused on the PVD neuron, which elaborates a branching dendrite arbor from two primary dendrites that run laterally along the animal, as well as a single axon that extends ventrally to make presynaptic connections in the ventral nerve cord (VNC), thereby providing a useful system for visualizing multiple distinct processes (*Figure 1A*). Expressing *ptrn-1a(cDNA)::tdTomato* in a subset of cells including the PVD neuron, we observed irregularly spaced puncta of PTRN-1a::tdTomato throughout the PVD dendrites and axon (*Figure 1B,C*, *Figure 1—figure supplement 3A,B*). A similar punctate localization was observed from PTRN-1 tagged with GFP at the N-terminus or with YFP at the C-terminus. We often observed continuous PTRN-1::tdTomato fluorescence in the primary dendrite adjacent to the cell body, and fewer, farther spaced puncta in the quaternary processes (*Figure 1B* and data not shown).

We also examined PTRN-1a::tdTomato localization in the PHC sensory neuron, which has the simple bipolar morphology more typical of *C. elegans* neurons (*Figure 3A*). In the PHC dendrite, PTRN-1::tdTomato localized to puncta with spacing similar to that of the PVD neuron (*Figure 1—figure supplement 3C*), suggestive that the PTRN-1a localization pattern in neurites is similar across different neuron classes.

We used live imaging to examine the dynamics of the PTRN-1::tdTomato puncta in the PVD neurites (*Video 1*, *Video 2*, *Figure 1—figure supplement 3D,E*). In 40-min videos, the majority of the PTRN-1::tdTomato puncta exhibit some slow movement. The puncta can be seen dividing, appearing and growing, dissolving, and merging. The movements of each punctum are not obviously correlated with those of the surrounding puncta.

## PTRN-1 stabilizes MT foci in neuronal processes and body wall muscle

To examine whether PTRN-1 binds MTs in neurites, we co-expressed PTRN-1a::tdTomato with EMTB::GFP, the MT-binding domain of ensconsin fused to GFP (*Masson and Kreis, 1993*; *Bulinski and Bossler, 1994*; *Faire et al., 1999*). EMTB::GFP, which binds dynamically along the side of MTs, has been previously used to visualize MTs in vivo (*Bulinski et al., 2001*; *Lechler and Fuchs, 2007*; *von Dassow et al., 2009*; *Wühr et al., 2010*). In *C. elegans* neurons, it generally exhibited continuous fluorescence throughout neuronal processes. In the PVD neuron, the dendritic arbor consists of processes that branch perpendicularly from each other, getting progressively thinner with each branching event (*Albeg et al., 2011*). Accordingly, EMTB::GFP fluorescence in the PVD neuron was strong in the primary dendrites but weak and sometimes discontinuous in the tertiary and quaternary dendrites, likely because these narrow neurites contain few MTs (*Figure 1D*, *Figure 1—figure supplement 4A,B*) (*Albeg et al., 2011*).

We assessed the relationship between MTs and PTRN-1 by examining the fluorescence of these two fusion proteins in the body wall muscle cells, which have larger cell bodies than neurons, providing more space in which MTs are organized. EMTB::GFP-labeled MTs were strung throughout the cytosol of these cells (*Figure 1F*, *Figure 1—figure supplement 4E,F*). They also formed parallel lines along the sarcolemma, from which emanated rows of evenly-spaced puncta visible in the next confocal slice or two closer to the membrane (slices were 0.4 µm apart) (*Figure 1E*, *Figure 1—figure supplement 4C,D*). These puncta appeared to be MT ends. Hence, the disorganized MT strands are

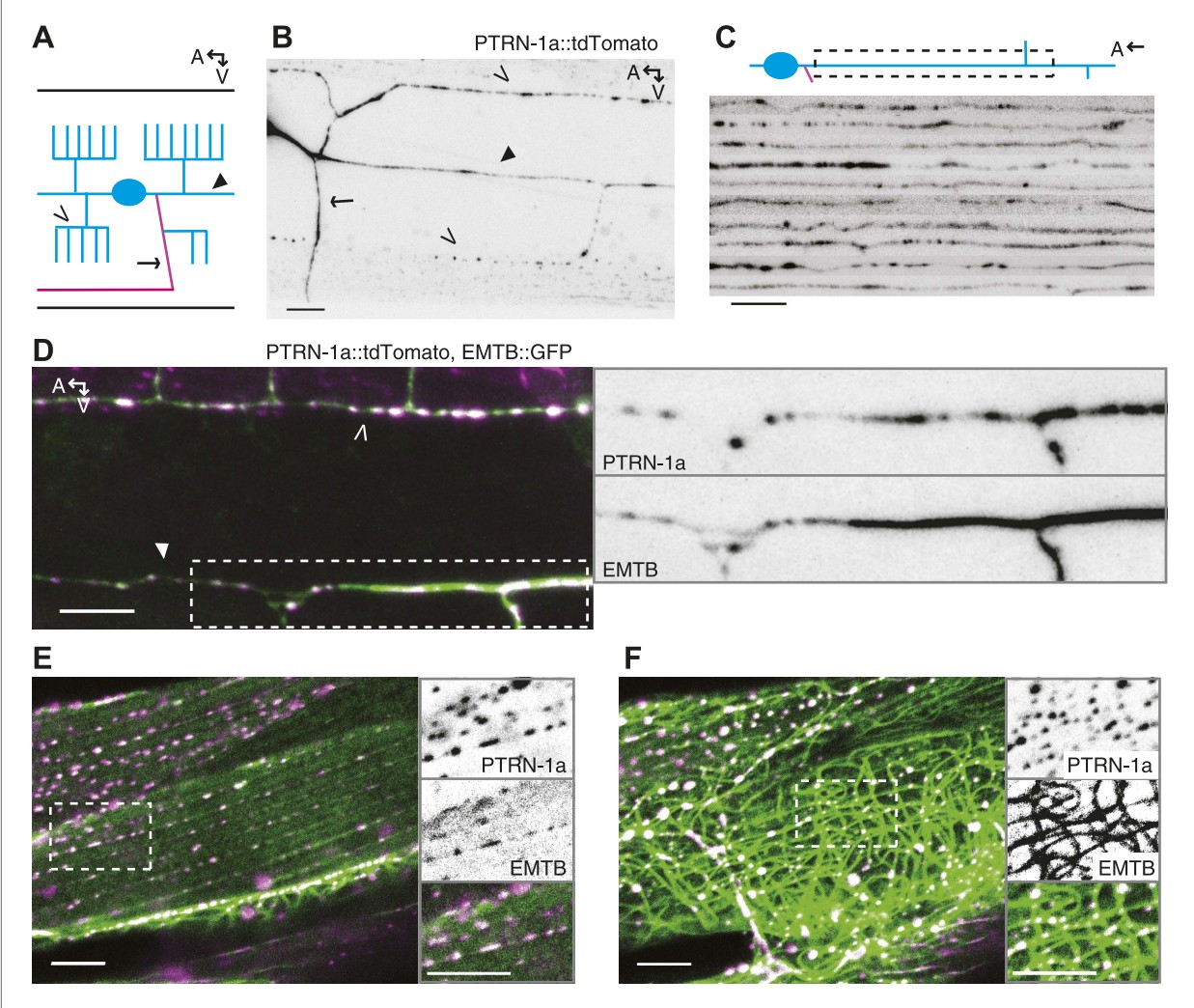

**Figure 1**. PTRN-1 localizes to puncta throughout neurites and colocalizes with MTs. (**A**) Schematic diagram of the central region of the PVD neuron. The cell body (blue oval) is in the posterior half of the animal. An elaborate dendritic arbor (blue lines) extends from the base of the head to the posterior of the animal, and the single axon (magenta) is extended into the ventral nerve chord (VNC). Black lines represent the outline of the animal. (**B**) PTRN-1a::tdTomato localization in the PVD neuron. The cell body is outside of the image, close to the left edge. (**C**) Confocal micrographs from 10 animals showing PTRN-1a::tdTomato localization in the PVD primary dendrite directly posterior to the cell body. (**D–F**). Colocalization of PTRN 1a::tdTomato (magenta) and EMTB::GFP (green) in the PVD neurites (**D**), at the sarcolemma of the body wall muscle cells (**E**), and in the cell interior of the body wall muscle cells (**F**). Data were acquired from *wyEx5968* and *wyEx6022* transgenes coexpressed in the *ptrn-1(tm5597)* mutant. Closed arrowhead indicates the primary dendrite, the open arrowheads indicate tertiary dendrite, and arrow points to axon of the PVD neuron (**A**, **B**, **D**). A, anterior; V, ventral. Scale bar: 5 μm.

The following figure supplements are available for figure 1:

**Figure supplement 1**. PTRN-1 is broadly expressed.

**Figure supplement 2**. PTRN-1 exhibits punctate localization in neuronal processes and the body wall muscle cells.

**Figure supplement 3**. PTRN-1 localizes to puncta throughout the neurites in the PVD and PHC neurons.

**Figure supplement 4**. EMTB::GFP binds MTs in the PVD neuron and the body wall muscles.

**Figure supplement 5**. Highly expressed PTRN-1a::tdTomato binds along MT filaments.

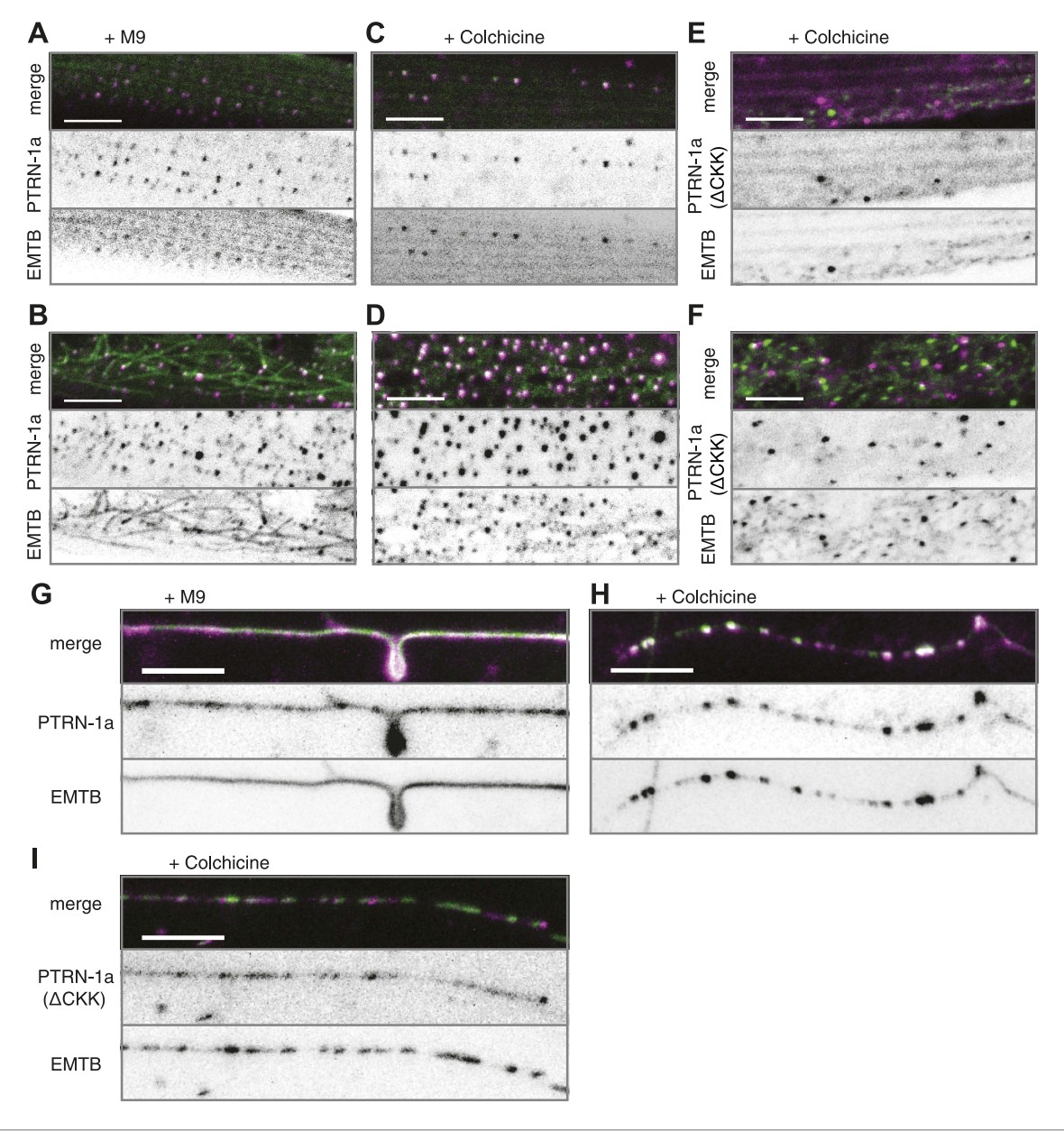

**Figure 2**. PTRN-1 stabilizes MT foci in neurons and muscles. (**A**–**D**) PTRN-1a::tdTomato and EMTB::GFP at the sarcolemma (**A** and **C**) and cell interior (**B** and **D**) of body wall muscle cells after acute colchicine exposure (**C** and **D**) or M9 control (**A** and **B**). (**E** and **F**) PTRN-1a(ΔCKK)::tdTomato and EMTB::GFP at the sarcolemma (**E**) and cell interior (**F**) of body wall muscle cells after acute colchicine exposure. (**G**–**H**) Localization of PTRN-1a::tdTomato and EMTB::GFP in the PVD dendrite after acute colchicine exposure (**H**) or M9 control (**G**). (**I**) PTRN-1a(ΔCKK)::tdTomato and EMTB::GFP in the PVD primary dendrite after acute colchicine exposure. All data acquired from *wyEx5968* with either *wyEx6022* (**A**–**D** and **G** and **H**), *wyEx6092* (**I**), or *wyEx6165* (**E** and **F**) co-expressed in *bus-17(e2800)* mutant animals. Scale bar: 5 μm.

The following figure supplements are available for figure 2:

**Figure supplement 1**. Acute colchicine exposure changes EMTB::GFP localization in body wall muscles.

**Figure supplement 2**. PTRN-1::tdTomato colocalizes with EMTB::GFP puncta after MT depolymerization by colchicine.

**Figure supplement 3**. PTRN-1a(ΔCKK) exhibits punctate localization in body wall muscle cells and neurons.

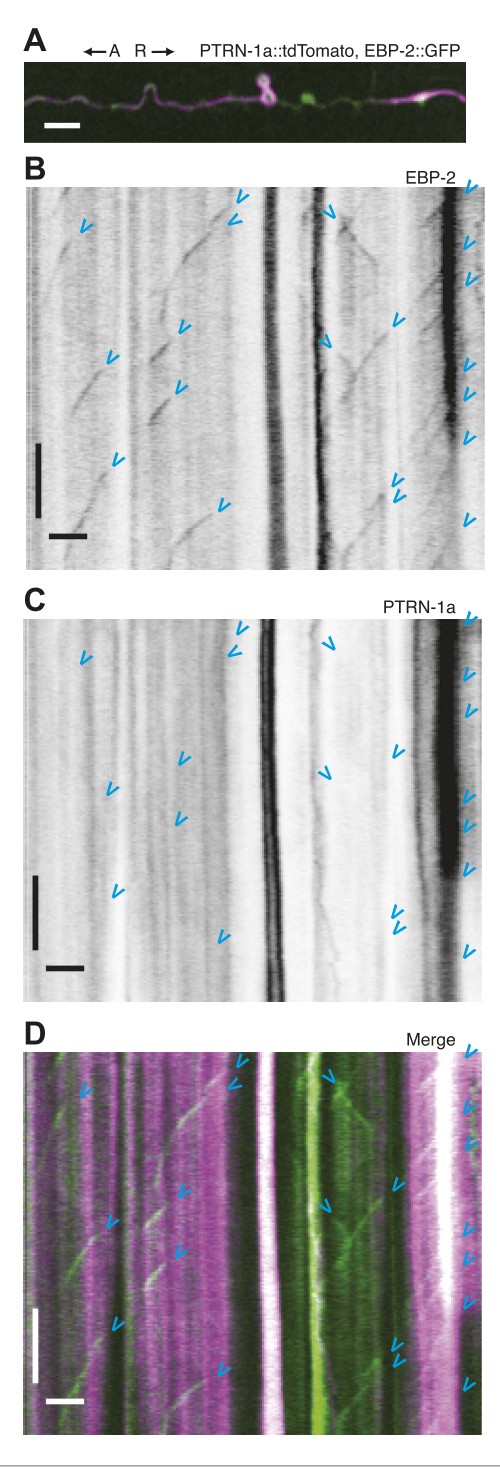

**Figure 3**. Immobility of PTRN-1a::tdTomato puncta contrasts with EBP-2::GFP movements in the PVD dendrite. (**A**) Live imaging was performed on a *ptrn-1 (tm5597)* mutant animal co-expressing EBP-2::GFP (green, from the *wyEx4828* transgene), which labels growing plus-end of MTs, and PTRN-1a::tdTomato (magenta, from the *wyEx6022* transgene) along a section of the tertiary dendrite of the PVD neuron. (**B**–**D**). Kymographs of EBP-2::GFP (**B**),

*Figure 3. Continued on next page*

anchored at the regularly spaced loci on or near the plasma membrane. The angle and spacing of the MT lines at the sarcolemma suggest that they run parallel and perhaps adjacent to the dense bodies (the *C. elegans* equivalent of Z-discs) and M-lines. This microtubule organization in the body wall muscle cells resembles the pattern observed by fluorescence-labeled ELP-1, the *C. elegans* EMAP (Echinoderm Microtubule-Associated Protein)-like protein (*Hueston et al., 2008*). In mammalian muscle fibers, MTs filaments form both a grid-like organization aligned with the Z-discs, which is dependent on dystrophin (*Prins et al., 2009*), and squiggles in the cytosol with less apparent organization (*Ralston et al., 1999*).

Whether fused to YFP or tdTomato, PTRN-1 localized to evenly spaced puncta at the sarcolemma and to irregularly spaced puncta throughout the interior of the body wall muscle cells (*Figure 1—figure supplement 2A–C*, *Figure 1E,F*). The PTRN-1a::tdTomato puncta within the muscle cytosol always co-localized with the EMTB::GFP cytosolic threads (*Figure 1F*), confirming that PTRN-1 localizes to MTs. Further, PTRN-1a puncta colocalized with EMTB::GFP puncta at the sarcolemma (*Figure 1E*), a finding which suggests that, like its homologs in fruitflies and humans (*Meng et al., 2008*; *Goodwin and Vale, 2010*; *Tanaka et al., 2012*), PTRN-1a localizes to MT ends. It is unclear how PTRN-1::tdTomato is localized at either the sarcolemma or in the muscle cell interior. Interestingly, in mammalian muscle, MTs are nucleated from the immobile Golgi elements strung throughout the cytoplasm (*Oddoux et al., 2013*).

Although CAMSAP proteins preferentially bind to the minus ends of MTs, when they are overexpressed, CAMSAPs have also been shown to bind along the side of MTs (*Meng et al., 2008*; *Baines et al., 2009*; *Goodwin and Vale, 2010*). Similarly, highly expressed PTRN-1a::tdTomato localized along the side of MTs in the body wall muscle cells (*Figure 1F* (bottom left of main panel), *Figure 1—figure supplement 5*).

We next sought to determine whether PTRN-1 localizes to sites where MTs are stabilized. We treated animals with the MT depolymerizing drug colchicine for 1 hr and examined the effect on EMTB::GFP localization. Because the *C. elegans* cuticle is largely impermeable to colchicine, we performed this experiment in the *bus-17(e2800)* genetic background, which has increased permeability to drugs, including colchicine (*Leung et al., 2008*; *Gravato-Nobre et al., 2005*; *Bounoutas et al., 2009*). Acute colchicine

*Figure 3. Continued*

PTRN-1a::tdTomato (**C**), and overlay of EBP-2::GFP (green) with PTRN-1a::dtTomato (magenta) (**D**) from a 110 s video acquired from the PVD process shown in **A**. Time runs top to bottom. Arrows point to start of EBP-2::GFP movements. A, anterograde; R, retrograde. Scale bar: 5 μm, ~22 s.

The following figure supplements are available for figure 3:

**Figure supplement 1**. Immobility of PTRN-1a::tdTomato puncta contrasts with EBP-2::GFP movements in the PVD dendrite.

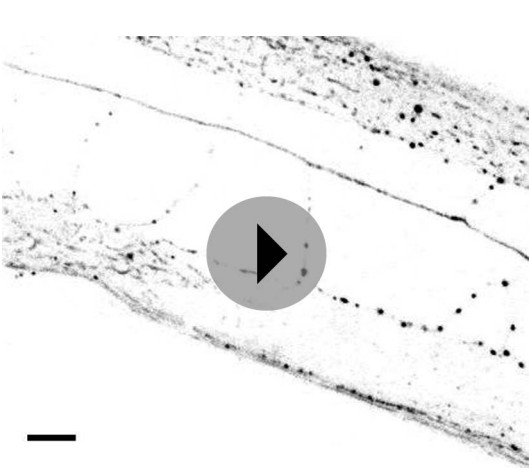

**Video 1**. PTRN-1::tdTomato movements in the PVD neuron. First example video of a L4 animal showing 40 min with 90 s/frame. See *Figure 1—figure supplement 3D* for diagram of neuron morphology.

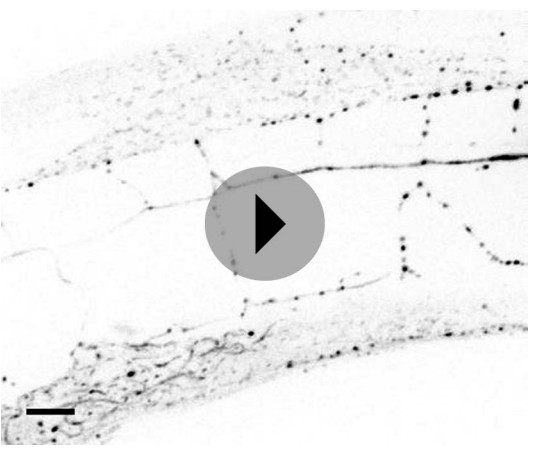

**Video 2**. PTRN-1::tdTomato movements in the PVD neuron. Second example video of a L4 animal showing 40 min with 90 s/frame. See *Figure 1—figure supplement 3E* for diagram of neuron morphology.

treatment dramatically altered the distribution of EMTB::GFP such that fibers were no longer visible. Small EMTB::GFP foci remained at the cell membrane and in the cell interior, along with a haze of fluorescence throughout the cytosol (*Figure 2—figure supplement 1A,B*). Since EMTB::GFP normally binds to the sidewalls of MTs (*Faire et al., 1999*), this change in its distribution confirms that the colchicine treatment led to MT depolymerization, as expected. The localization of PTRN-1a::tdTomato appeared to be unaffected by the acute colchicine exposure, and the PTRN-1a::tdTomato puncta co-localized with the EMTB::GFP puncta both at the sarcolemma and in the cell interior (*Figure 2A–D*, *Figure 2—figure supplement 2*). These data show that, in the body wall muscle cells, the localization of PTRN-1a puncta is not dependent on MTs. They further imply that PTRN-1a localizes to sites where MTs are stabilized, even under conditions that cause complete depolymerization of all other MTs in the cell. Finally, because PTRN-1a::tdTomato and EMTB::GFP puncta exhibit a regular, repeating pattern at the sarcolemma, the fact that these rows are unaffected by the acute colchicine treatment indicates that these puncta represent sites of MT anchorage.

We used acute colchicine treatment to examine whether PTRN-1a likewise localizes to sites of MT stabilization in neurons. In the PVD neuron, acute colchicine exposure caused the continuous EMTB::GFP staining to dissolve into closely spaced puncta (*Figure 2G,H*). As in the body wall muscle cells, the PTRN-1a::tdTomato localization in the PVD neurites appeared unaffected by the MT depolymerization, and the remaining EMTB::GFP colocalized with the PTRN-1a::tdTomato puncta (*Figure 2G,H*, *Figure 2—figure supplement 2*). We interpret these data as suggestive that PTRN-1a localizes to sites where MTs are stabilized in the neurites.

As previous studies have shown that the CKK domain of CAMSAP proteins is involved in MT binding (*Baines et al., 2009*; *Goodwin and Vale, 2010*), we analyzed PTRN-1a(ΔCKK)::tdTomato to determine whether PTRN-1a itself stabilizes MTs. PTRN-1a(ΔCKK)::tdTomato exhibited similar localization in body wall muscle cells as full-length PTRN-1a::tdTomato, though these PTRN-1a(ΔCKK)::tdTomato puncta sometimes did not colocalize with EMTB::GFP (*Figure 2—figure supplement 3A,B*).

Performing acute colchicine treatment on animals co-expressing EMTB::GFP and PTRN-1a(ΔCKK)::tdTomato in the body wall muscle cells, we found that GFP-stained MT filaments

were transformed into GFP puncta, but these puncta did not colocalize with PTRN-1a(ΔCKK)::tdTomato (*Figure 2E,F*, *Figure 2—figure supplement 2*). There appeared to be fewer puncta in the muscle cells after the acute colchicine treatment, particularly at the sarcolemma. This may indicate that PTRN-1a(ΔCKK)::tdTomato localization in the muscle cells is dependent on MTs. The EMTB::GFP foci present after acute colchicine treatment in this strain might be stabilized by endogenous PTRN-1 and/or other MT binding proteins.

Finally, although PTRN-1a(ΔCKK)::tdTomato localized to puncta in the PVD processes (*Figure 2—figure supplement 3C*), after acute colchicine exposure, PTRN-1a(ΔCKK)::tdTomato puncta exhibited reduced colocalization with EMTB::GFP (*Figure 2I*, *Figure 2—figure supplement 2*). Taken together, these data indicate that one of the MT ends is more stable than the rest of the MT in vivo, possibly due to end-binding proteins, and PTRN-1a::tdTomato itself stabilizes MT foci.

## PTRN-1 supports MT polymerization initiation in neuronal processes

Live imaging of EBP-2 (EB1), an MT-binding protein that specifically associates with growing plus ends, has been used to visualize MT polymerization in *C. elegans* neurites (*Mimori-Kiyosue et al., 2000*; *Maniar et al., 2012*). To examine the relationship between PTRN-1 and dynamic MT plus ends, we performed live imaging on PVD tertiary dendritic processes co-expressing EBP-2::GFP with PTRN-1a::tdTomato (*Figure 3*, *Figure 3—figure supplement 1*). As reported for other neurons, each EBP-2::GFP punctum appeared, migrated in a single direction, and disappeared (blue arrowheads, *Figure 3B*, *Figure 3—figure supplement 1B,E*). In some cases, multiple EBP-2::GFP movements emanated from the same position in the course of a video, suggestive of a local factor that promotes MT polymerization from these loci. There were also motionless EBP-2::GFP puncta that were present whether the neuron expressed the EBP-2::GFP transgene alone or with the PTRN-1a::tdTomato (*Figure 3B*, *Figure 3—figure supplement 1C,F,I–K*).

PTRN-1a::tdTomato, in contrast, was localized almost exclusively to immobile puncta for the duration of these 110 s videos (*Figure 3C*, *Figure 3—figure supplement 1C,G*). Many of the EBP-2::GFP movements appeared to emanate from PTRN-1a::tdTomato puncta, though the close spacing of the PTRN-1a::tdTomato puncta makes quantification of this observation impracticable (*Figure 3D*, *Figure 3—figure supplement 1D,H*).

To investigate the requirement for PTRN-1 in neurite MT dynamics, we performed live imaging of EBP-2::GFP movements in wild-type vs *ptrn-1* mutant animals. We obtained two *ptrn-1(null)* alleles: *tm5597*, which carries an intragenic deletion that introduces an early nonsense mutation, and *wy560*, a 65 kb deletion that spans the entire *ptrn-1* locus (*Spilker et al., 2012*). Strains carrying either of the *ptrn-1(null)* alleles exhibited grossly wild-type growth, development, and neuronal morphology (*Figure 1—figure supplement 1*, *Figure 4—figure supplement 1*, and data not shown).

As the PHC dendrite has been used previously to monitor EBP-2::GFP movements (*Yan et al., 2013*), we used this system to examine EBP-2::GFP movements in the *ptrn-1* mutants (*Figure 4A*). In wild-type animals, EBP-2::GFP comets move predominantly toward the cell body (*Yan et al., 2013*), consistent with the known minus-end-out polarity of MTs in dendrites (*Burton, 1985*). In both of the *ptrn-1* mutant strains, we observed fewer total EBP-2::GFP movements than in the wild-type strain (*Figure 4B*), but the direction of EBP-2::GFP movements was like that of wild-type (*Figure 4C*). Expression of *ptrn-1a::tdTomato* in the PHC neuron of *ptrn-1(tm5597)* mutant animals rescued the decreased number of EBP-2::GFP movements (*Figure 4B*), indicating that the requirement for PTRN-1 in promoting EBP-2::GFP movements is cell-autonomous. These data implicate PTRN-1 in promoting MT polymerization in the dendrite but not directly organizing MT polarity. This loss of dynamic MTs in the *ptrn-1* mutants could be indicative of a reduction in the total number of MTs in the neurite, which would be suggestive of a role for PTRN-1 in MT nucleation or stabilization. These data do not quantify the stable MT population, however, so an alternate explanation for the reduction in EBP-2::GFP movements is that there is an increase in neurite MT stability in the *ptrn-1* mutants.

## PTRN-1 promotes MTs stability in neurites

To determine whether PTRN-1 promotes neurite MT stabilization, we examined the interaction between *ptrn-1* and the MT destabilizing drug colchicine. Although the impenetrability of the *C. elegans* cuticle at the L4 and older stages necessitated the use of the *bus-17* mutation for the acute colchicine treatment described above, wild-type animals reared from hatching in a low dose of colchicine exhibit defects in MT organization and neuronal function, indicating that the drug reaches the neurons in this

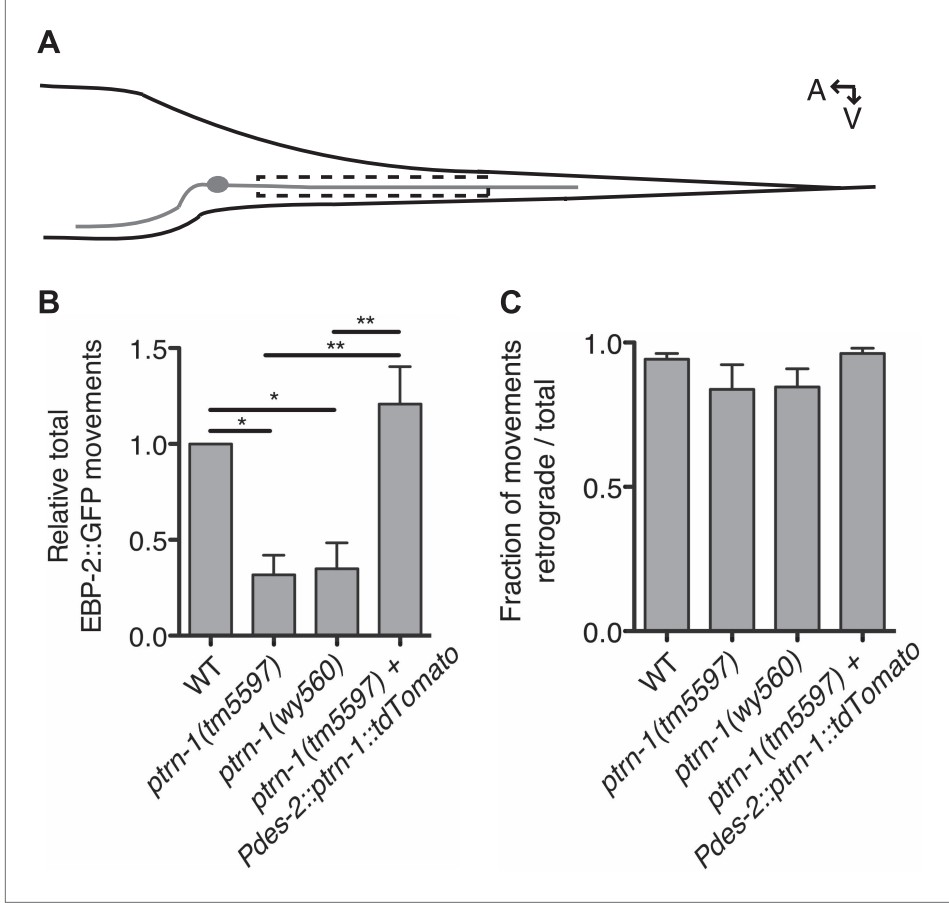

**Figure 4**. PTRN-1 promotes MT polymerization in neurites. (**A**) Schematic diagram of the PHC neuron. The anterior process is the axon; the posterior process is the dendrite. Live imaging was used to monitor EBP-2::GFP movements in the boxed region of the PHC dendrite. A: anterior, V: ventral. (**B** and **C**) Quantification of EBP-2::GFP anterograde and retrograde movements in the PHC dendrite of wild-type (WT) vs *ptrn-1(tm5597)* and *ptrn-1(wy560)* mutant animals, and vs the *ptrn-1(tm5597)* mutant carrying the *Pdes-2::ptrn-1::tdTomato* transgene, which is expressed in a subset of neurons as well as the body wall muscle. (**B**) Total EBP-2::GFP movements in each strain normalized against the wild-type control. (**C**) Fraction of EBP-2::GFP movements in each strain that moved in the retrograde direction. Mean ± SEM. (n = 3 experiments, each with at least 10 animals/genotype, *p<0.05, **p<0.01, ANOVA with Bonferroni post test).

The following figure supplements are available for figure 4:

**Figure supplement 1**. Neuronal morphology is grossly unaffected by loss of *ptrn-1*.

longer timeframe (***Chalfie and Thomson, 1982***). Furthermore, rearing animals in a low dose of colchicine has been shown to suppress neurite morphology defects caused by several dominant alleles of β-tubulin *mec-7*, supporting the hypothesis that these alleles caused increased MT stability (***Savage et al., 1994***; ***Kirszenblat et al., 2013***), Therefore, this method of administering colchicine can reveal pharmacogenetic interactions between colchicine and genes that affect neuronal MT stability.

Although the neurite morphology of many *C. elegans* neurons in the *ptrn-1(tm5597)* mutant all appeared grossly wild-type under normal growth conditions (***Figure 4—figure supplement 1***), growth in a low dose of colchicine caused dramatic ectopic sprouting from the sides of neurites in the *ptrn-1 (tm5597)* mutant but not wild-type animals (***Figure 5***). In the DD/VD-type motorneurons, the cell bodies are situated in the ventral nerve cord (VNC), and a single unbranched commissure per cell connects processes in the VNC with those in the dorsal nerve cord. In wild-type animals grown in the presence of colchicine, these processes generally appear to be largely morphologically normal (***Figure 5A,C***). In the *ptrn-1(tm5597)* mutant grown in colchicine, in contrast, we observed ectopic branching of the

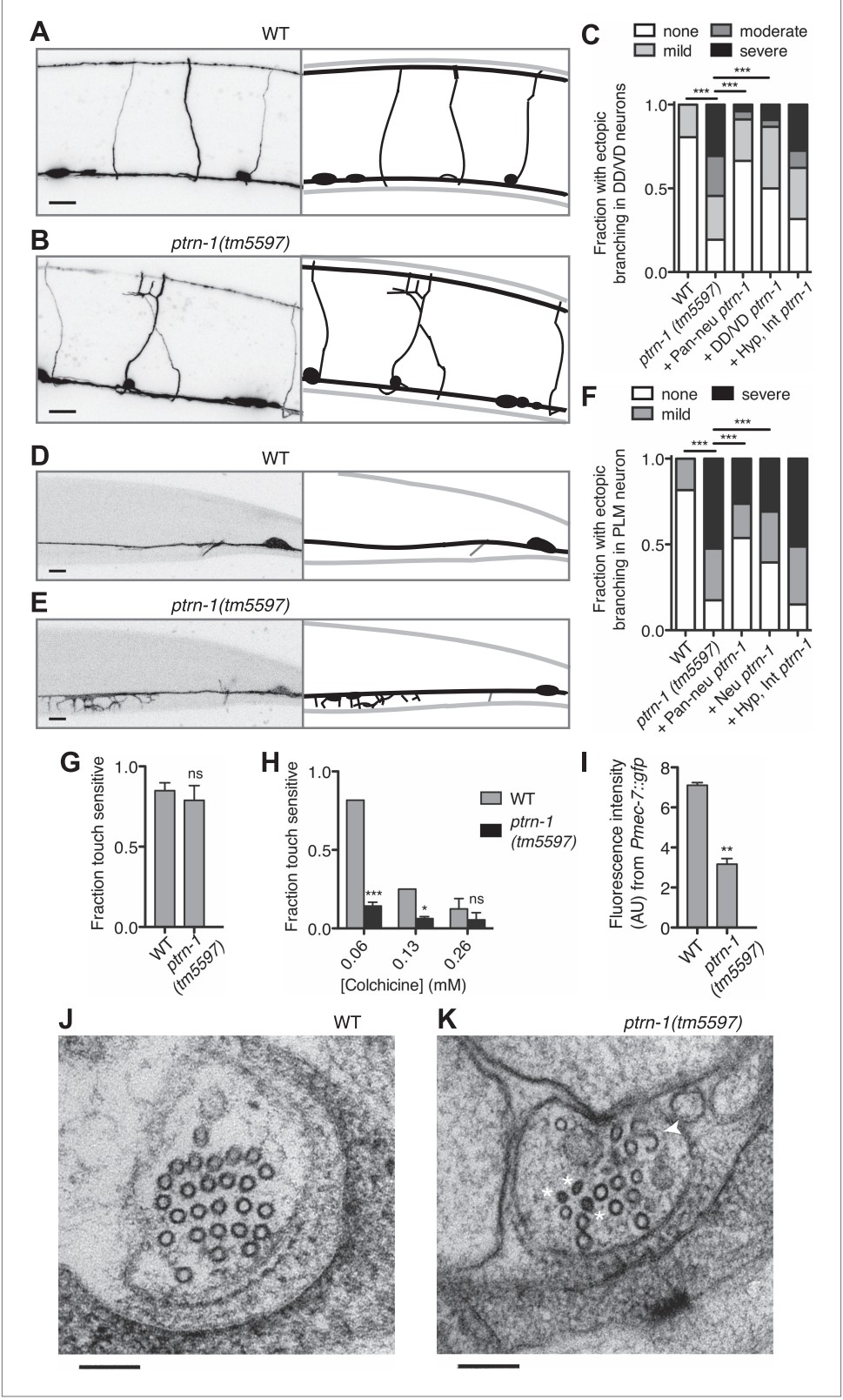

**Figure 5**. PTRN-1 supports MT stability in neurites. (**A-C**) Wild-type (**A**) and *ptrn-1(tm5597)* mutant (**B**) animals were grown in 0.13 mM colchicine to the L4 stage, and cytosolic RFP was used to visualize the DD/VD neurons. Scale bar: 10 μm. (**C**) Fraction of animals with ectopic sprouting from the DD/VD neurons, scored based on severity as

*Figure 5. Continued on next page*

*Figure 5. Continued*

described in 'Materials and methods' (n = at least 80 animals/genotype, ***p<0.001, Chi-squared test with Šidák correction). (**D–F**) Wild-type (**D**) and *ptrn-1(tm5597)* mutant (**E**) animals were grown in 0.035 mM colchicine to the L4 stage, and the PLM neuron was visualized with myr::GFP. Scale bar: 5 μm. (**F**) Fraction of animals exhibiting ectopic sprouting from the PLM neuron, scored based on severity (n = at least 60 animals/genotype, ***p<0.001, Chi-squared test with Šidák correction). In schematic diagrams, the light gray lines represent the outline of the animal, DD and PLM neurons are black, and other neurons (in **D** and **E** only, one short, unbranched process near the PLM cell body of each image) are dark gray. For tissue specific rescue, DD/VD (*Punc-47L*), Pan-neu: pan-neuronal (*Prab-3*), Hyp: hypodermal (*Pdpy-7*), Int: intestinal (*Pvha-6*), neu: a subset of neurons including PLM (*Punc-86*). (**G**) Touch sensitivity of wild-type vs *ptrn-1(tm5597)* mutant animals. Mean ± SEM. (n = 3 experiments, each with 8–12 animals/genotype, ns not significant (p=0.46), *t* test). (**H**) Touch sensitivity of wild-type vs *ptrn-1(tm5597)* mutant animals grown in the indicated concentrations of colchicine. Mean ± SEM. (n = 2 experiments, each with 10 animals/genotype, ***p<0.001, *p<0.05, *t* test for each drug concentration). (**I**) Average fluorescence of GFP expressed from the *Pmec-7* (β-tubulin) promoter in the PLM cell body of wild-type vs *ptrn-1* mutant animals. Mean ± SEM. (n = 2 experiments, each with at least 13 animals/genotype, **p<0.01, *t* test). (**J** and **K**) Transmission electron microscopy of the PLM neuron in wild-type (**J**) and *ptrn-1(tm5597)* mutant (**K**) young adult animals, sectioned near the rectum. Note MTs with abnormally smaller diameters (Asterisks), and a MT sheet shaped like an 'S' (Arrow head). Scale bar: 100 nm.

The following figure supplements are available for figure 5:

**Figure supplement 1**. PTRN-1 protects the ALM touch receptor neuron against ectopic neurite sprouting during growth in colchicine.

commissures, as well as additional neurites sprouting from processes in the VNC (*Figure 5B,C*). This ectopic branching could be rescued by tissue-specific *ptrn-1* expression either pan-neuronally or exclusively in the DD/VD neurons (*Figure 5C*). This synthetic interaction between colchicine and *ptrn-1* is suggestive that *ptrn-1* promotes MT stabilization in the DD/VD-type neurons.

Of note, we also tested the hypothesis that the colchicine-induced branching in the *ptrn-1* mutant could be due to increased in vivo colchicine levels compared to wild-type. The most likely cause of such an effect would be *ptrn-1*-dependent defects in the hypodermis or intestine, tissues involved in drug uptake in *C. elegans* (*Leung et al., 2008*). We therefore created transgenic *ptrn-1* mutants expressing PTRN-1a in the hypodermis and intestine. This transgene had little or no effect on colchicine-induced ectopic branching in the DD/VD neurons (*Figure 5C*), indicating that the drug–gene interaction leading to ectopic neuronal branching is unlikely to be due to increased drug permeability.

Whereas most *C. elegans* neurons contain four to six 11-protofilament (pf) MTs, the six touch receptor neurons (TRNs) have a strikingly different MT organization. These mechanoreceptor neurons have 15-pf MTs produced from tubulin genes expressed predominantly in the TRNs (*Chalfie and Thomson, 1979*; *Hamelin et al., 1992*; *Fukushige et al., 1999*). The TRNs have 25–50 MTs per neurite cross-section in young adult animals. Both the morphology and function of these cells are particularly sensitive to perturbations in MT stability (*Chalfie and Thomson, 1982*). We focused on the two PLM neurons, which each elaborate both an anterior-directed process and a posterior-directed process from the cell bodies located at the base of the tail (*Figure 4—figure supplement 1G,H*). A single commissure extends from each of the PLM anterior processes to the VNC, where PLM makes presynaptic connections with other neurons, making these anterior-directed processes axon-like. The posterior-directed processes make neither presynaptic nor postsynaptic connections.

Under normal growth conditions, the morphology of the PLM neurons in the *ptrn-1(tm5597)* mutant was largely wild-type (*Figure 4—figure supplement 1*), with several more subtle defects described below. Growth in a low dose of colchicine, however, resulted in extensive ectopic sprouting from the PLM axon in the *ptrn-1(tm5597)* mutant but not in the wild-type strain (*Figure 5D–F*). Similar ectopic sprouting was observed in the ALM neuron, another TRN in the anterior half of the animal (*Figure 5—figure supplement 1*). Tissue-specific *ptrn-1* expression either in all neurons or in subset of neurons that includes PLM but not in both the hypodermis and the intestine rescued this ectopic sprouting of the touch receptor neurons (*Figure 5F*).

The PLM axon is extended during embryogenesis. As the *C. elegans* eggshell is impermeable to colchicine (*Bounoutas et al., 2009*), the ectopic sprouting occurs after neurogenesis has been completed. Therefore, the ectopic sprouting reflects a defect in neurite maintenance rather than neurite

outgrowth during development. Taken together, these data indicate that the loss of *ptrn-1* enhances sensitivity to colchicine cell-autonomously in neurons containing either 11-pf or 15-pf MTs. This enhanced sensitivity to colchicine likely reflects reduced MT stability in the *ptrn-1* mutant, suggesting that PTRN-1 promotes MT stabilization.

TRNs mediate the behavioral response to light touch (*Chalfie, 2009*). We assessed the functionality of the TRNs in the absence of *ptrn-1* function by quantifying light touch sensitivity in *ptrn-1(tm5597)* mutant vs wild-type animals. We found no significant difference in light touch response between the *ptrn-1(tm5597)* mutant and wild-type animals grown in the absence of colchicine, though our assay may have lacked sufficient sensitivity to parse subtle differences (*Figure 5G*). Growing the animals in several concentrations of colchicine, however, we found that the *ptrn-1(tm5597)* mutant lost light touch sensitivity at a lower concentration of colchicine than the wild-type strain (*Figure 5H*).

MT destabilization has long been known to negatively regulate tubulin production (*Cleveland, 1988*). In the *C. elegans* TRNs, MT destabilization induced by genetic or pharmacological manipulations results in not only decreased levels β-tubulin *mec-7* mRNA but also a general decrease in protein production, including GFP driven by the *mec-7* promoter (*Savage et al., 1994*; *Bounoutas et al., 2011*). Similarly, the *ptrn-1(tm5597)* mutant exhibited decreased GFP fluorescence from a *Pmec-7::gfp* transgene relative to the wild-type strain (*Figure 5I*), further implicating PTRN-1 in promoting MT stability.

MT density and protofilament composition in the neurites of the TRNs have been well characterized by electron microscopy (*Chalfie and Thomson, 1979, 1982*; *Cueva et al., 2007*; *Cueva et al., 2012*). To better understand the function of PTRN-1, we used electron microscopy to compare the PLM MTs in the *ptrn-1(tm5597)* mutant to wild-type. In cross sections of the PLM neuron in wild-type animals reared at 25°C, there are 25–50 15-pf MTs and occasionally one or two 11-pf MTs (*Figure 5J*) (*Chalfie and Thomson, 1979*; *Cueva et al., 2012*; our unpublished data). We examined cross sections of the PLML/R axons from two *ptrn-1(tm5597)* mutant young adult animals sectioned at the rectum and found they had 13, 15, 2, and 14 MTs, respectively. The majority of these MTs had the characteristically large diameter of 15-pf MTs, but several MTs had smaller diameters indicative of 11-pf MTs (*Figure 5K*). Furthermore, in one of the PLML cross sections, we observed an irregular MT structure that persisted through three serial sections (150 nm) that was 'S' shaped instead of circular (*Figure 5K*). We speculate that such a structure might have formed from two circular MTs opening and then joining. The reduction in MT number found in all four cells, as well as the 'S' shaped MT structure in one, implicates a role for PTRN-1 in maintaining the integrity of MTs in neuronal processes.

## PTRN-1 is required for proper neurite morphology and synaptic material localization in the PLM neuron

Given the requirement for *ptrn-1* in MT stability in the PLM neuron, we examined the effect of *ptrn-1* deficiency on PLM morphology in greater detail. Roughly 20% of *ptrn-1(tm5597)* mutant L4 animals exhibited defective extension of the PLM commissure (*Figure 6A–C*). In wild-type animals, this commissure is extended during the L1 larval stage from the axon to the VNC posterior of and close to the vulva, and it is present in every L4 animal (*Figure 6A*). In *ptrn-1(tm5597)* mutants with this defect, we generally observed one or several ventrally directed buds along the region of PLM axon, where the commissure is normally positioned (*Figure 6B*). The defective PLM commissure extension observed in the *ptrn-1(tm5597)* mutant was fully rescued by *ptrn-1a* cDNA expressed in the PLM neuron, indicating that the role for PTRN-1 in commissure formation is cell-autonomous (*Figure 6C*).

We next examined the localization of synaptic material in the PLM neuron. Each PLM neuron has a presynaptic specialization in the VNC at the end of the commissure, where synaptic vesicles (SVs) and associated proteins such as the small GTPase RAB-3 and SNB-1/synaptobrevin are localized (*Chalfie et al., 1985*; *Schaefer et al., 2000*). In wild-type animals, mCherry::RAB-3 localized to two patches along the VNC that correspond to the synaptic patches of the two PLM neurons (*Figure 6D,H*). In the *ptrn-1* mutants, there was an incompletely penetrant loss of mCherry::RAB-3 at the synaptic patch region, and in many *ptrn-1* mutant animals, at least one of the PLM neurons had no visible accumulation of mCherry::RAB-3 at the synaptic patch (*Figure 6E,H*). The *ptrn-1(wy560)* strain had a higher penetrance of this defect than the *ptrn-1(tm5597)* strain (*Figure 6H*). The *ptrn-1(wy560)* allele is a deletion that removes not only the entire *ptrn-1* locus but also seven surrounding ORFs, including the Muscleblind homolog *mbl-1*. As *mbl-1* has been shown to promote the accumulation of synaptic material at the presynaptic region of other *C. elegans* neurons (*Spilker et al., 2012*), we speculate that this difference in penetrance might be due to *mbl-1* deficiency in the *wy560* allele.

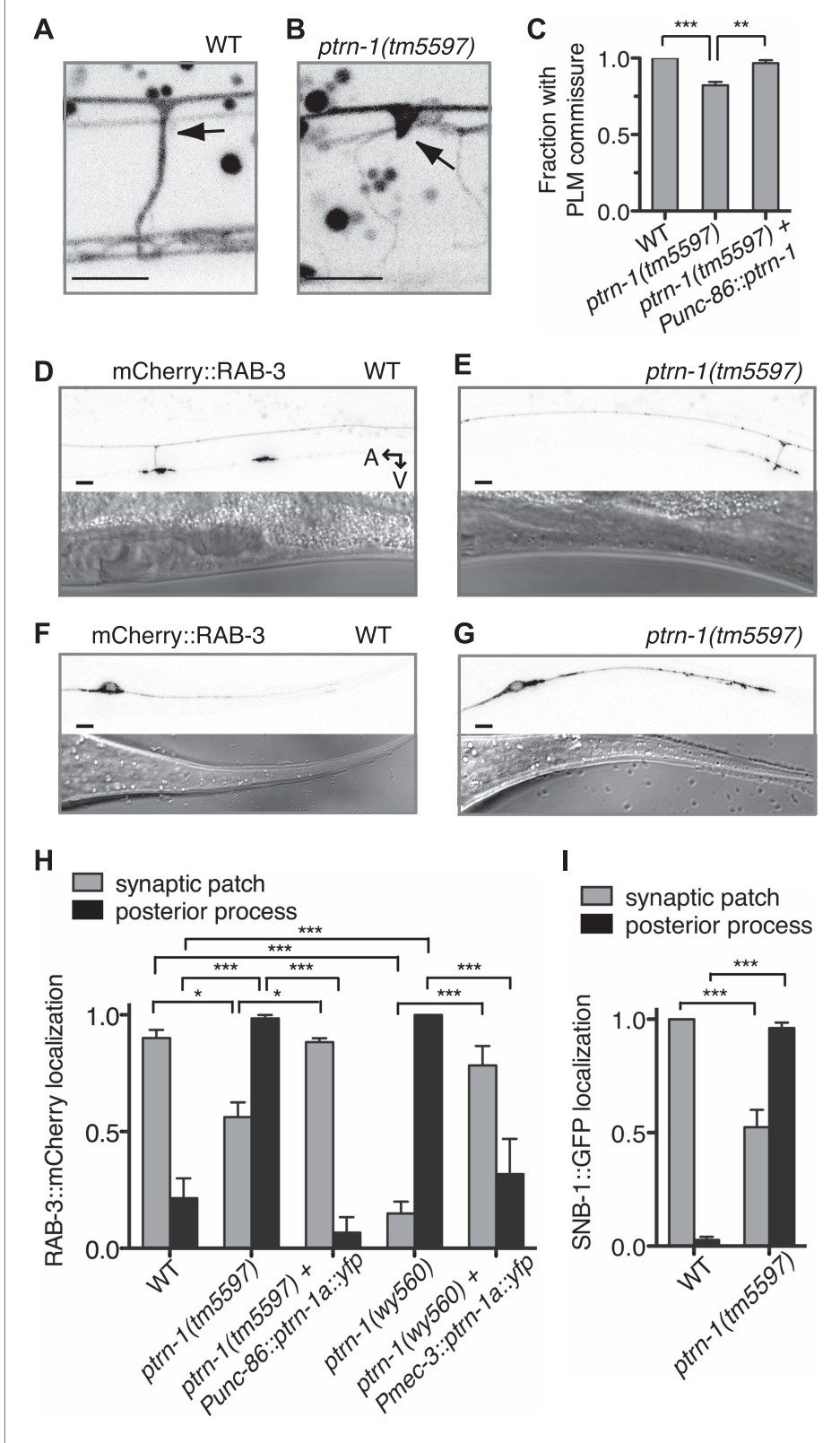

**Figure 6**. PTRN-1 promotes synapse localization and neurite morphology in the PLM neuron. (**A** and **B**) myrGFP was used to visualize the PLM commissure in wild-type (**A**) and *ptrn-1(tm5597)* mutant (**B**) animals. Arrows point to commissure or commissure bud. (**C**) Fraction of animals with a PLM commissure connecting the axon to the ventral

*Figure 6. Continued on next page*

*Figure 6. Continued*

nerve cord. Mean ± SEM. (n = 3 experiments, each with at least 30 animals/genotype, **p<0.01, ***p<0.001, one-way ANOVA with Bonferroni post test). (**D** and **E**) mCherry::RAB-3 at the synaptic patch of the PLM neurons of wild-type (**D**) and *ptrn-1(tm5597)* mutant (**E**) animals. (**F** and **G**) mCherry::RAB-3 in the posterior process of the PLM neurons in wild-type (**F**) and *ptrn-1(tm5597)* mutant (**G**) animals. (**H**) Fraction of wild-type and *ptrn-1* mutant animals with visible accumulation of mCherry:RAB-3 at the synaptic patch and the posterior process of the PLM neuron. The *Punc-86* promoter is expressed in a subset of neurons including the TRNs; the *Pmec-3* promoter is expressed in the TRNs. Animals with two visible mCherry:RAB-3 patches in the PLM synaptic region were counted as having synaptic accumulation, and animals with one or no visible mCherry::RAB-3 patches were considered to have loss of synaptic accumulation. Mean ± SEM. (n = 2 experiments, each with 30 animals/genotype, *p<0.05, ***p<0.001, two-way ANOVA with Bonferroni post test). (**I**) Fraction of wild-type and *ptrn-1(tm5597)* mutant animals with visible accumulation of SNB-1::GFP at the synaptic patch and the posterior process of the PLM neuron. Synaptic patch accumulation was scored as in H. Mean ± SEM. (n = 2 experiments, each with at least 20 animals/trial. ***p<0.001, two-way ANOVA with Bonferroni post test).

In addition to the loss of mCherry::RAB-3 from the synaptic patches, we observed a fully penetrant ectopic accumulation of mCherry::RAB-3 in the PLM posterior process in both *ptrn-1* mutant strains (*Figure 6F–H*). The localization of SNB-1::GFP in the PLM neuron of *ptrn-1(tm5597)* was similar to that of RAB-3::mCherry (*Figure 6I*). These data implicate a requirement for *ptrn-1* in proper SV localization.

To determine whether the requirement for *ptrn-1* in SV localization is cell-autonomous, we used two different promoters to drive *ptrn-1a::yfp* cDNA expression in the PLM neuron of the *ptrn-1* mutants. Both constructs rescued the defects in mCherry::RAB-3 localization (*Figure 6H*), indicating that PTRN-1 functions cell autonomously in the PLM neuron to promote proper SV localization.

What mechanism underlies the aberrant commissure formation and SV mislocalization in the *ptrn-1* mutant? DLK-1 is a conserved mitogen-activated protein kinase kinase kinase (MAPKKK) that functions in a variety of situations in neurons, including neurite outgrowth, synapse development, and axon regeneration (*Tedeschi and Bradke, 2013*). In *C. elegans*, the DLK-1 pathway is required to mediate the response to MT destabilization in the PLM neuron, and it also promotes proper synapse localization (*Nakata et al., 2005*; *Bounoutas et al., 2011*). Indeed, hyperactivation of the DLK-1 pathway causes a defect in the PLM commissure similar to that observed in the *ptrn-1(tm5597)* mutant, albeit with a higher penetrance (*Grill et al., 2007*). We examined PLM commissure formation and SV localization in a *dlk-1(ju476); ptrn-1(tm5597)* double mutant. In this double mutant, we found that *dlk-1* completely suppressed both the commissure extension defect and the loss of mCherry::RAB-3 from the synaptic patch (*Figure 7A*), indicating that *dlk-1* is required to mediate these aspects of the *ptrn-1* mutant phenotype. The PMK-3 p38 MAPK functions downstream of DLK-1 (*Nakata et al., 2005*). We observed similar suppression of these *ptrn-1* phenotypes in a *pmk-3(ok169); ptrn-1(tm5597)* double mutant (*Figure 7—figure supplement 1*), corroborating the role for the *dlk-1* pathway in mediating the *ptrn-1* PLM commissure formation and SV localization defects.

We next asked whether the synthetic interaction between *ptrn-1* and colchicine that resulted in neurite sprouting from the PLM axon is also mediated through the DLK-1 pathway. Indeed, *dlk-1 (ju476)* completely suppressed the neurite sprouting observed in *ptrn-1(tm5597)* during growth on colchicine (*Figure 7B–D*). Interestingly, the wild-type strain reared in a higher dose of colchicine exhibited neurite sprouting similar to that seen in the *ptrn-1(tm5597)* mutant at the lower colchicine concentration, and this sprouting was likewise suppressed by *dlk-1(ju476)* (*Figure 7B,E,F*). Similar effects of *dlk-1* were observed on colchicine-induced neurite sprouting in the ALM neurons (data not shown).

DLK-1 was not required to mediate all of the abnormal phenotypes caused by *ptrn-1* loss of function, however: the ectopic accumulation of mCherry::RAB-3 in the PLM posterior process was similar in the *dlk-1(ju476); ptrn-1(tm5597)* double mutant to that of the *ptrn-1(tm5597)* mutant strain (*Figure 7A*). Therefore, the accumulation of SVs in the PLM posterior process is separable from the loss of SVs at the synaptic patch, and it is mediated by a mechanism other than the DLK-1 pathway.

Finally, we asked whether the reduction in EBP-2::GFP movements in the *ptrn-1* mutant is also dependent on *dlk-1*. Interestingly, we observed an increase of roughly two-fold in EBP-2::GFP movements in the PHC dendrite in the *dlk-1(ju476)* single mutant relative to wild-type (*Figure 7G*). Loss

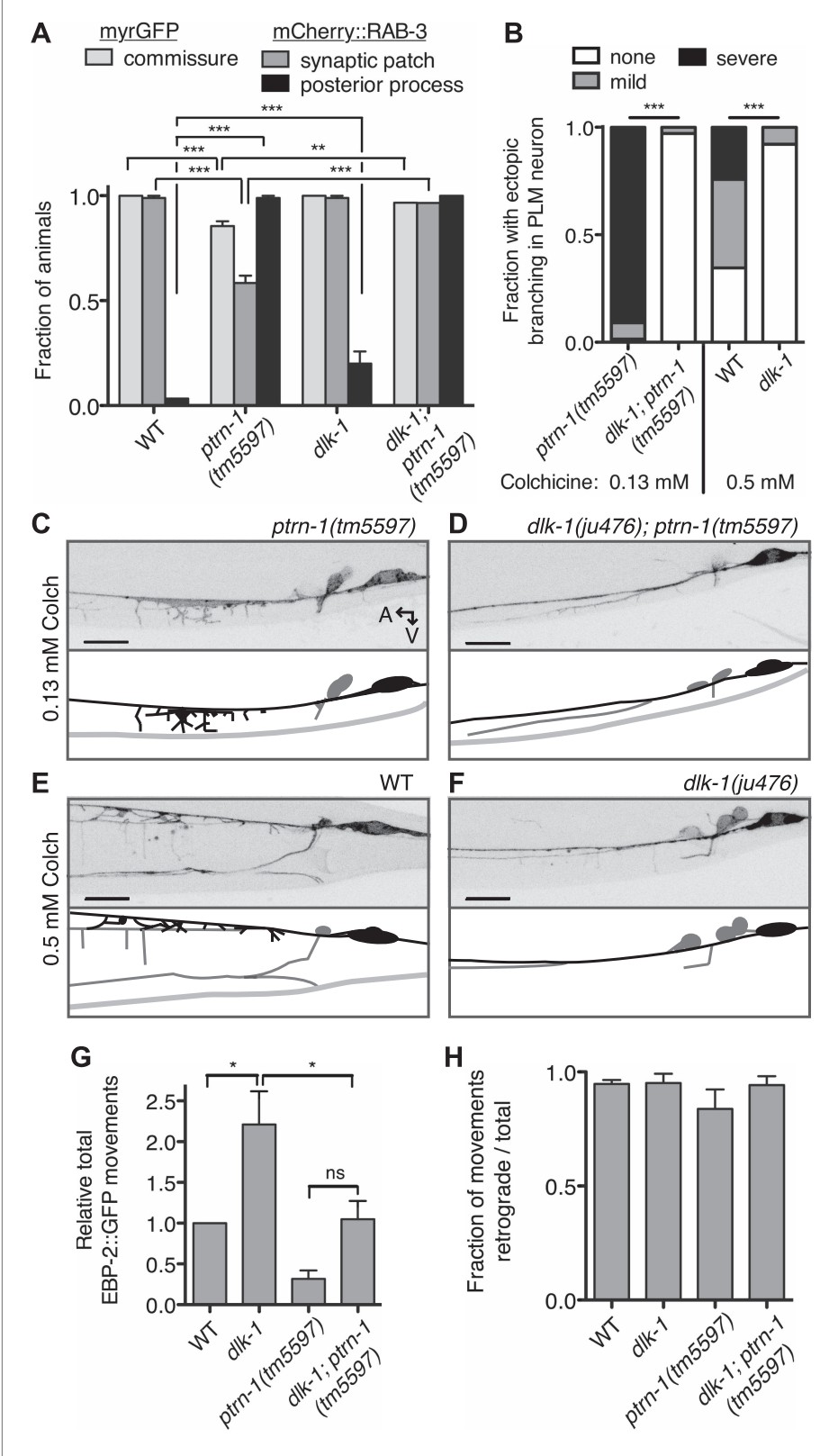

**Figure 7**. Aberrant phenotype of the PLM neuron in the *ptrn-1* mutant is mediated partially by the DLK-1 pathway. (**A**) The *wyIs97(Punc-86::myrGFP, Punc-86::mCherry::rab-3)* transgene was used to simultaneously visualize commissure formation and SV localization. Only animals with an intact PLM commissure were counted for *Figure 7. Continued on next page*

*Figure 7. Continued*

mCherry:RAB-3 localization at the synaptic patch. Values represent mean ± SEM. (n = 3 experiments, each with at least 30 animals/genotype, **p<0.01, ***p<0.001, two-way ANOVA with Bonferroni post test). (**B–F**) Animals were grown in 0.13 mM colchicine (**C** and **D**) or 0.5 mM colchicine (**E** and **F**) to the L4 stage, and the PLM neuron was visualized with myr::GFP. In schematic diagrams, the PLM neurons are black, and other neurons in the image are dark gray. (**B**) Fraction of animals exhibiting ectopic sprouting from the PLM neuron, scored based on severity (n = at least 120 animals/genotype, ***p<0.001, Chi-squared test for each drug concentration). (**G** and **H**) Quantification of EBP-2::GFP anterograde and retrograde movements in the PHC dendrite. (**G**) Total EBP-2::GFP movements in each strain normalized against the wild-type control. (**H**) Fraction of EBP-2::GFP movements in each strain that moved in the retrograde direction. Mean ± SEM. (n = 3 experiments, each with at least 9 animals/genotype, *p<0.05, ns not significant, one-way ANOVA with Bonferroni post test. The data for the *ptrn-1* single mutant is the same as that shown in ***Figure 4***).

The following figure supplements are available for figure 7:

**Figure supplement 1**. Loss of *pmk-3* partially suppresses the aberrant phenotype of the PLM neuron in the *ptrn-1* mutant.

of *dlk-1* had no effect on the orientation of EBP-2 movements (***Figure 7H***). There was a trend for the *dlk-1(ju476); ptrn-1(tm5597)* double mutant to have increased EBP-2::GFP movements relative to the *ptrn-1* single mutant, though this difference was not statistically significant (***Figure 7G***). However, the *dlk-1(ju476); ptrn-1(tm5597)* double mutant exhibited fewer movements than the *dlk-1* single mutant (***Figure 7G***). This intermediate phenotype in the *dlk-1(ju476); ptrn-1(tm5597)* double mutant indicates that the DLK-1 pathway is not the only mechanism required for the *ptrn-1* mutant phenotypes, and it is suggestive that DLK-1 functions partially in parallel to PTRN-1 to influence EBP-2::GFP movements in the PHC neuron.

## Discussion

The mechanisms preventing depolymerization from the MT minus ends within neuronal processes are a long-standing mystery. Previous studies have shown that CAMSAP family proteins directly bind MT minus ends (***Meng et al., 2008***; ***Goodwin and Vale, 2010***). Our data suggest that PTRN-1 binds to MT minus ends and protects them from depolymerization in neuronal processes. Multiple lines of evidence support this conclusion. First, PTRN-1 localized to puncta throughout the neuronal processes that directly bind and stabilize MTs. Of note, electron microscopy reconstruction of neurites in the VNC showed that MT ends are staggered, with an average distance between minus ends of roughly 1.7 µm (***Chalfie and Thomson, 1979***). Second, our electron microscopy data revealed that the *ptrn-1 (tm5597)* mutant strain had fewer total MTs and some MTs with abnormal structure in the PLM neuron. Third, the *ptrn-1* mutant exhibited a pharmacogenetic enhancement with colchicine in respect to neurite morphology. Fourth, the *ptrn-1* mutant exhibited a reduction in the number of neurite MT polymerization events as determined by counting EBP-2::GFP movements. Finally, the SV mislocalization and aberrant neurite branching observed in the *ptrn-1* mutant were suppressed by *dlk-1*, which is known to influence the effects of MT destabilization in *C. elegans.*

The *D. melanogaster* CAMSAP protein Patronin localizes to MT minus ends throughout the cytoplasm in interphase S2 cells (***Goodwin and Vale, 2010***). Acute MT depolymerization in this system resulted in puncta of mCherry-tubulin that co-localized with the GFP–Patronin foci, similar to our findings in *C. elegans* neurons and muscle cells in vivo. Importantly, by allowing MT repolymerization, Goodwin and Vale established that the GFP–Patronin foci represented MT nucleation centers. It is unclear from our studies whether PTRN-1 localizes to MT nucleation sites in neurites, though the colocalization between PTRN-1 and the beginning of EBP-2 movements is suggestive that it might be.

*H. sapiens* CAMSAP3 (Nezha) was originally identified as a component of epithelial cell adherens junctions, where it is anchored by cadherins and p120-catenin (***Meng et al., 2008***). We showed that PTRN-1 puncta in neuronal processes were unaffected by drug-induced MT depolymerization, and PTRN-1 localization was independent of the CKK domain thought to bind MTs (***Baines et al., 2009***; ***Goodwin and Vale, 2010***). These data suggest that, like CAMSAP3 at adherens junctions, PTRN-1 is localized in an MT-independent manner.

If PTRN-1 were the sole mechanism protecting minus ends in neurites, we would expect the phenotype of mutants carrying *ptrn-1* null alleles to include severe defects in neuronal morphogenesis. We observed, however, relatively mild defects under standard growth conditions. Because of these data, we speculate that PTRN-1 functions in parallel with other mechanisms that promote MT stability in *C. elegans* neurites. These are likely to include tubulin posttranslational modifications and MT-associated proteins (*Poulain and Sobel, 2010*). Of particular interest, tubulin detyrosination protects MTs from depolymerization by kinesin-13 family motors in fibroblasts and neurites (*Peris et al., 2009*; *Ghosh-Roy et al., 2012*). Because *D. melanogaster* Patronin protects MTs from kinesin-13-mediated depolymerization (*Goodwin and Vale, 2010*; *Wang et al., 2013*), these modifications are an attractive candidate for how MTs are stabilized in the absence of *ptrn-1*. Defective regulation of α-tubulin acetylation, another prevalent posttranslational tubulin modification in neurites, causes abnormal neurite morphology and function in both mice and *C. elegans* (*Creppe et al., 2009*; *Topalidou et al., 2012*), though its effect on MT stability is uncertain and may be circumstance-dependent (*Janke and Kneussel, 2010*). In *C. elegans,* electron microscopy studies have shown that the loss of α-tubulin acetyltransferases causes a decrease in MT abundance, increase in MTs with irregular protofilament number, and appearance of MTs in which the protofilament lattice had opened into semicircular or sheet-like structures (*Cueva et al., 2012*; *Topalidou et al., 2012*). Further, the loss of the α-tubulin acetyltransferase MEC-17 in the PLM neuron causes an increase in dynamic MTs and, in older adult animals, loss of synaptic material at the synaptic region accompanied by accumulation of synaptic material in the posterior process (*Neumann and Hilliard, 2014*).

Consistent with the notion that *ptrn-1* functions in parallel with other mechanisms to promote MT stability, the *ptrn-1* mutant strain grown in a low dose of colchicine exhibited aberrant neurite outgrowth. A higher dose of colchicine caused similar ectopic branching in the PLM neuron of the wild-type strain, indicating that MT destabilization is sufficient to elicit this phenotype. What is the mechanism by which MT destabilization leads to the neurite outgrowth from axons? When collateral branches form along an axon, the budding branch and surrounding axon have fewer, shorter MTs than regions of the axon with no collateral branching (*Yu et al., 1994*). Further, pharmacological or genetic manipulations that decrease MT stability have been shown to cause neurite outgrowth along the length of mature neurites (*Bray et al., 1978*; *Yu et al., 2008*). Perhaps the combination of *ptrn-1* mutation with colchicine results in fewer, shorter MTs in these neurites, and this MT status promotes ectopic sprouting. Alternatively, the response to MT destabilization resulting from loss of PTRN-1 function may be more akin to the regeneration response to an axonal lesion, since this also results in ectopic branching and growth cone formation in motor and sensory neurons (*Hammarlund et al., 2009*; *Yan et al., 2009*).

Taken together, our data indicate that PTRN-1 represents one of the elusive factors that stabilize the MT minus ends in neurites, promoting both the stable and dynamic MTs during development and maintenance of the nervous system. Through regulation of MTs, PTRN-1 supports proper SV localization and the balance between neurite stability and remodeling.

## Materials and methods

### Nematode strains and maintenance

*C. elegans* strains were cultured on *E. coli* OP50 as described (*Brenner, 1974*). Data were collected from L4 stage animals except where otherwise noted, and all experiments were performed at 25°C because this elevated temperature enhanced the neuronal defects in the *ptrn-1* mutants (data not shown), except for those shown in *Figure 7B–F*, which were performed at 20°C. The *ptrn-1(tm5597)* allele was obtained from the National Bioresource Project in Japan and backcrossed three times. The *ptrn-1(wy560)* was isolated from RB771 provided by the CGC (*Spilker et al., 2012*), which is funded by NIH Office of Research Infrastructure Programs (P40 OD010440). The following additional strains were used in this study: N2 Bristol, TV13426 *ptrn-1(tm5597)*, TV15320 *ptrn-1(tm5597); wyEx6181* [*ptrn-1::gfp::SL2::mCherry* in fosmid WRM0615Ab03; *Podr-1::gfp*], TV14056 *ptrn-1(wy560); wyEx5730* [*Pptrn-1::ptrn-1a::yfp; Podr-1::gfp*], TV15195 *ptrn-1(tm5597); wyEx6022* [*Pdes-2::ptrn-1a::tdTomato; Podr-1::gfp*], TV15770 *ptrn-1(tm5597); wyEx6022; wyEx5968* [*Pdes-2::EMTB::gfp; Podr-1::rfp*], TV15399 *ptrn-1(tm5997); wyEx6216* [*Pmyo-3::ptrn-1a::tdTomato; rol-6(d)*], TV15773 *ptrn-1(tm5597); wyEx6165* [*Pmyo-3::ptrn-1a(ΔCKK)::tdTomato; rol-6(d)*]; *wyEx5968*, TV15790 *ptrn-1(tm5597); wyEx6092* [*Pdes-2 ::ptrn-1a(ΔCKK)::tdTomato; Podr-1::gfp; Pdes-2::bfp*]; *wyEx5968*, TV14687 *wyEx5968*, TV15383 *ptrn-1*

*(tm5597); wyEx5968, ptrn-1(tm5597); [Punc-86::gfp::ptrn-1a; Podr-1::rfp], TV15774 bus-17(e2800); wyEx5968, bus-17(e2800); wyEx6022; wyEx5968, TV15772 bus-17(e2800); wyEx6092; wyEx5968, TV15776 bus-17(e2800); wyEx6165; wyEx5968, ptrn-1(tm5597); wyEx4828 [Pdes-2::ebp-2::gfp; Podr-1::gfp]; wyEx6022, TV11781 wyEx4828, TV14069 ptrn-1(tm5597); wyEx4828, TV13424 ptrn-1(wy560); wyEx4828, TV16422 ptrn-1(tm5597); wyIs602; wyEx4828, TV12310 wyIs371 [ser-2prom3::myrGFP, Prab-3::mCherry, Podr-1::rfp], TV15768 wyIs371; ptrn-1(tm5597), TV1204 wyIs75 [Pexp-1::gfp, Punc-47L::rfp], TV15151 wyIs75; ptrn-1(tm5597), TV15314 wyEx6177 [pPD117.01 Pmec-7::gfp, Podr-1::gfp], TV15317 ptrn-1(tm5597); wyEx6177, TV1838 wyIs97 [Punc-86::myr-gfp, Punc-86::mCherry::rab-3], TV13422 wyIs97;ptrn-1(tm5597), TV12134 wyIs348[Pmec-17::mCherry::rab-3, Pmec-17::CD4::spGFP1-10], TV13423 wyIs348; ptrn-1(tm5597), TV15322 wyIs348; ptrn-1(tm5597); wyEx6023 [Punc-86::gfp::ptrn-1], TV13430 wyIs348; ptrn-1(wy560), TV14063 wyIs348; ptrn-1(wy560); wyEx5782 [Pmec-3::ptrn-1::yfp], NM0664 jsIs37 [Pmec-7::snb-1::gfp], TV14346 jsIs37; ptrn-1(tm5597), TV15777 dlk-1(ju476); wyIs97, TV15778 dlk-1(ju476); wyIs97; ptrn-1(tm5597), TV16093 pmk-3(ok169) wyIs97, TV16240 pmk-3(ok169) wyIs97; ptrn-1(tm5597), TV16396 wyIs75; wyEx6577 [Podr-1::gfp], TV16394 wyIs75; ptrn-1(tm5597); wyEx6575 [Podr-1::gfp], TV16398 wyIs75; ptrn-1(tm5597); wyEx6579 [Prab-3::ptrn-1a, Podr-1::gfp line 1], TV16399 wyIs75; ptrn-1(tm5597); wyEx6580 [Prab-3::ptrn-1a, Podr-1::gfp line 2], TV16395 wyIs75; ptrn-1(tm5597); wyEx6576 [Punc-47L::ptrn-1, Podr-1::gfp line 1], TV16397 wyIs75; ptrn-1(tm5597); wyEx6578 [Punc-47L::ptrn-1, Podr-1::gfp line 2], wyIs75; ptrn-1(tm5597); [Pdpy-7::ptrn-1, Pvha-6::ptrn-1, Podr-1::gfp lines 1-4], wyIs97; wyEx6582 [Podr-1::rfp], TV16401 wyIs97; ptrn-1(tm5597); wyEx6582 [Podr-1::gfp], TV16400 wyIs97; ptrn-1(tm5597); wyEx6589 [Prab-3::ptrn-1, Podr-1::rfp], TV16402 wyIs97; ptrn-1(tm5597); wyEx6583 [Punc-86::ptrn-1, Podr-1::rfp], wyIs97; ptrn-1(tm5597); [Pdpy-7::ptrn-1, Pvha-6::ptrn-1, Podr-1::rfp lines 1-4].*

## Molecular biology and transgenic lines

Expression vectors were made in the pSM vector, a derivative of pPD49.26 (A Fire, unpublished data) with added cloning sites (S McCarroll and CI Bargmann, unpublished data) using standard techniques. Plasmids were coinjected with markers *Podr-1::gfp*, *Podr-1::rfp*, or *rol-6(d)*.

## Confocal imaging and fluorescence microscopy

Images were acquired with a Zeiss LSM510 confocal microscope using a Plan-Apochromat 63x/1.4 objective. Data were analyzed using ImageJ software. Visual inspection and quantification of the penetrance of fluorescence localization were performed using a Zeiss Axioplan 2 microscope with a 63x/1.4NA objective and Chroma HQ filter sets for GFP, YFP, and RFP. Animals were immobilized in 2.5 mM levamisol +0.225 mM BDM (2,3-butanedione monoxime) or 2 mM levamisole for confocal imaging or fluorescence microscopy, respectively (Sigma, St Louis, MO). Colocalization was assessed using the Colocalization Finder in ImageJ.

## Dynamic imaging

Dynamic imaging was performed on an inverted Zeiss Azio Observer Z1 microscope using a Plan-Apochromat 63x/1.4 objective. L4 stage animals were anesthetized in 0.1% tricane +0.01% tetramisole for 15–30 min, then mounted on a 5% agarose pad on a slide for imaging (*Sulston and Horvitz, 1977*). All videos were acquired with a Quantum 512C camera. Videos of animals co-labeled with EBP-2::GFP and PQN-34a::tdTomato were 110-s videos with roughly 2 frames per second. Videos used to quantify EBP-2::GFP movements were 25-s videos with 8 frames per second. Kymographs were generated with ImageJ. *Video 1* and *Video 2* were acquired over 40 min with 90 s/frame. Z-stacks were acquired at each time point and maximum projections are shown.

## Colchicine treatment

For prolonged colchicine treatment, animals were grown from eggs on NGM plates containing colchicine as described (*Chalfie and Thomson, 1982*). For acute colchicine treatment, L4-stage animals were soaked in a drop of 10 mM colchicine in M9 or M9 alone for 1 hr. Animals were alive and thrashing at the end of the treatment.

## Mechanosensory assays

Mechanosensory assays were performed as described (*Hobert et al., 1999*). Briefly, L4 animals were tapped 10 times each with an eyelash, alternating between the anterior and posterior half of the body. The reaction was scored as either a 'response', if the animal reversed direction of movement,

or 'no response', if it did not. The fraction of touches resulting in a response were averaged for each animal to give the 'fraction touch sensitive'.

## Quantification of neurite ectopic branching

Animals from 3 to 6 plates/genotype were scored blind to genotype according to the following categories. For DD neurons, the categories were None: no ectopic branches or sprouting from any of the DD commissures or from the nerve cords, Mild: at least one branch or sprout from a commissure, Moderate: at least four ectopic branches, or three branches total from two different commissures, Severe: at least eight branching events, often with large growth cones projecting multiple filopodia. For the PLM axon, the categories were None: no ectopic branches, Mild: at least one branch or at least 4 large bulges along the axon, Severe: At least three branches, often many more accompanied by expanses of distended axon in the region containing the ectopic branches. For the tissue specific rescue experiments, four independent lines were initially assessed per each expression construct. We observed no rescue from any of the intestine-plus-hypodermis lines, and so all four lines were used for the experiments shown in *Figure 5C,F*. For the lines expressing *ptrn-1* in the neurons, we observed some lines provided stronger rescue than others, likely due to differences in expression level or mosaicism of the transgene. We therefore included the 1–3 lines with the most rescue per construct in the experiments shown in *Figure 5C,F*.

## Electron microscopy

Young adult wild-type N2 and *ptrn-1(tm5597)* animals were prepared as described (*Cueva et al., 2012*). Briefly, animals were frozen in an EMPACT2 high-pressure freezer system, and a Leica AFS freeze substitution apparatus (Vienna, Austria) was used to preserve in 2% glutaraldehyde plus 1% osmium tetroxide and embed in Epon/Araldite. A Leica Ultracut S microtome equipped with a diamond knife was used to cut 50-nm serial sections, which were collected on Formvar-coated copper slot grids. The grids were poststained to enhance contrast in 3.5% uranyl acetate (30 s) and Reynold's lead citrate preparation (3 min). The grids were imaged on a transmission electron microscope (JEOL TEM 1230, Tokyo, Japan), and images were acquired with an 11 megapixel bottom-mounted cooled CCD camera (Orius SC1000, Gatan, Pleasanton, CA).

## Acknowledgements

We thank the *Caenorhabditis* Genetics Center and the National Bioresource Project-Japan for strains. We thank EV Stewart for technical advice. This work was supported by the Howard Hughes Medical Institute and NIH.

## Additional information

### Funding

| Funder | Grant reference number | Author |
|---|---|---|
| Howard Hughes Medical Institute | | Kang Shen |
| National Institutes of Health | NS047715 | Miriam B Goodman |
| National Institutes of Health | EB006745 | Miriam B Goodman |
| National Institutes of Health | 1F32NS083135-01 | Claire E Richardson |

The funders had no role in study design, data collection and interpretation, or the decision to submit the work for publication.

### Author contributions

CER, Conception and design, Acquisition of data, Analysis and interpretation of data, Drafting or revising the article; KAS, Conception and design, Contributed unpublished essential data or reagents; JGC, Acquisition of data, Analysis and interpretation of data, Drafting or revising the article; JP, Acquisition of data, Analysis and interpretation of data; MBG, Analysis and interpretation of data, Drafting or revising the article; KS, Conception and design, Analysis and interpretation of data, Drafting or revising the article

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
