## [Decision Letter]

Thank you for sending your work entitled “PTRN-1, a microtubule minus end-binding CAMSAP homolog, promotes microtubule function in *C. elegans* neurons” for consideration at *eLife*.

Two reviewers were clearly more positive than a third reviewer but the positive reviewers maintained their essentially positive stance even after being made aware of the more negative comments. Therefore, we would like to invite you to submit a revised version of your manuscript in which we hope you can address a number of issues raised by the reviewers, as summarized below.

1) The conclusion that PRTN-1 protein is found at minus ends of microtubule rests to a large degree on work on patronin in other species; this needs to be clearly acknowledged throughout the manuscript.

2) The extensive fluorescence within the muscle cell needs to be explained. This fluorescence could presumably from ends of microtubules, but no information is given about the organization of microtubules within muscles in *C. elegans*.

3) It is stated that PTRN-1 binds to immobile puncta throughout the manuscript, but the authors look at most for less than two minutes. The authors need to look over longer times to assert that the puncta are immobile.

4) At several places in the manuscript the authors state that PTRN-1 stabilizes microtubules. Their main argument is that colchicine acts at a lower concentration. This result only says that the loss of this protein makes the microtubules more sensitive to the drug. Elsewhere the authors state that they see no difference in structure or number in microtubules except in the PLM. One would expect that the number of microtubules would go down. This needs to be explained.

5) The authors fail to provide statistics for their assertions. The note inadvertently left in the methods shows that at least one of the authors was aware of this problem.

6) A large part of the manuscript uses two novel fluorescent markers-labeled patronin and a labeled microtubule-binding domain. It would significantly strengthen the paper to validate these markers. Further, experiments using these markers (colocalization) are descriptive-some quantitative data should be provided.

7) The authors should more clearly define in the abstract and elsewhere which prtn-1 phenotypes are derived from the upregulation of DLK-1 signaling, i.e., which phenotypes are rescued by loss of DLK-1. DLK-1 is not mentioned in abstract or introduction, which gives the impression that the neurite morphology and synaptic vesicle location defects result directly from changes in MT structure caused by loss of PTRN-1. The authors state that PTRN-1 ‘anchors’ microtubules in neurons and muscle based on the observations that PTRN-1 puncta colocalize with MT puncta following colchicine treatment and that the PTRN-1 CKK deletion mutant often does not colocalize with MTs. As stated, this conclusion is perhaps too strong given that ptrn-1 mutants have a relatively minor, but significant, MT defect. I believe that the conclusion is reasonable, but that it should be stated more conservatively.

---

## [Author Response]

*1) The conclusion that PRTN-1 protein is found at minus ends of microtubule rests to a large degree on work on patronin in other species; this needs to be clearly acknowledged throughout the manuscript*.

The following alterations were made to the text to address this issue:

- To the Introduction, we added the following more thorough description of the previous work that established that CAMSAP proteins directly bind the MT minus end: “Meng et al. purified and fluorescence-labeled the C-terminal half of CAMSAP3 and sequentially added rhodamin-labeled and unlabeled MTs (39). The CAMSAP3 fragment colocalized with the end of the MT with higher rhodamine fluorescence, which indicates that it was bound to the minus end (39). Goodwin et al. found that purified GFP-Patronin which was attached to a coverslip bound and anchored rhodamine-MTs by a single end (25). Further, they used MT gliding assays in which either the plus-end motor kinesin or the minus-end motor dynein were added to the purified rhodamine-MT plus GFP-Patronin to show that the Patronin-bound end of the MT was the minus end (Goodwin et al., 2010).”

- The sentence “Further, the regularly spaced PTRN-1a::tdTomato puncta at the sarcolemma co-localized with the EMTB::GFP puncta at the sarcolemma (Figure 1), suggestive that PTRN-1a localizes to MT ends” was changed to “Further, PTRN-1a puncta colocalized with EMTB::GFP puncta at the sarcolemma (Figure 1), a finding which suggests that, like its homologs in fruitflies and humans ([39]; Goodwin et al. 2010; [55]), PTRN-1a localizes to MT ends”.

- The following was added to the Results:

“As previous studies have shown that the CKK domain of CAMSAP proteins is involved in MT binding ([4]; Goodwin et al., 2010), we analyzed PTRN- 1a(ΔCKK)::tdTomato to determine whether PTRN-1a itself stabilizes MTs.”

- The following sentence was added to the Discussion:

“Previous studies have shown that CAMSAP family proteins directly bind MT minus ends ([39]; Goodwin et al., 2010).”

- References were added as shown below in the Discussion:

“We showed that PTRN-1 puncta in neuronal processes were unaffected by drug-induced MT depolymerization, and PTRN-1 localization was independent of the CKK domain thought to bind MTs ([4]; Goodwin et al. 2010).”

*2) The extensive fluorescence within the muscle cell needs to be explained. This fluorescence could presumably from ends of microtubules, but no information is given about the organization of microtubules within muscles in* C. elegans*.*

There is little reported about the organization of MTs in *C. elegans* muscle, but the EMTB::GFP pattern we observe appears quite similar to the MT pattern shown by [31] using fluorescence-labeled ELP-1, the *C. elegans* EMAP-like homolog. We have added a reference to this paper. We have also added a brief description regarding MTs in mammalian muscle to the Results section.

The following sentences were added to the Results section, after the description of EMTB::GFP in the muscle cells, to better describe the known MT organization in muscle cells:

“This microtubule organization in the body wall muscle cells corroborates the pattern observed by fluorescence-labeled ELP-1, the *C. elegans* EMAP (Echinoderm Microtubule-Associated Protein)-like protein (31). In mammalian muscle fibers, MTs filaments form both a grid-like organization aligned with the Z-discs that is dependent on dystrophin (49) and squiggles in the cytosol with less apparent organization (50).”

And later:

“It is unclear how PTRN-1::tdTomato is localized at either the sarcolemma or in the muscle cell interior. Interestingly, in mammalian muscle, MTs are nucleated from the immobile Golgi elements strung throughout the cytoplasm (45).”

We use the muscle cells in this study because the large, three-dimensional cytoplasm makes the interaction between fluorescence-labeled PTRN-1 and MTs much more apparent than it is in the thin neurites. Because the import of this study is in describing the function of PTRN-1 in neurons, we prefer to keep the Discussion focused on this without additional speculation about the function of MTs and/or PTRN-1 in muscles.

*3) It is stated that PTRN-1 binds to immobile puncta throughout the manuscript, but the authors look at most for less than two minutes. The authors need to look over longer times to assert that the puncta are immobile*.

We followed this suggestion and performed live imaging on the PTRN-1::tdTomato puncta in the PVD neuron for 40 minutes. Interestingly, in these longer movies, although some of the PTRN-1::tdTomato puncta do not change their positions, the majority can be seen moving slowly. These slow movements of each PTRN-1::tdTomato punctum appear to be independent of the movements of the neighboring puncta, leading to some puncta merging with each other. Others can be seen dividing, and others disappear. We have incorporated two of these movies into the manuscript along with accompanying description in the text.

Although the PTRN-1::tdTomato puncta are stable compared to EBP-2::GFP, it seems from these movies that they are not*,* in fact, immobile over this longer timeframe. Before taking these movies, we had imagined that PTRN-1 might be part of a protein complex that linked MT minus ends to some anchor that attached the neurite to the surrounding tissue. Because of these movies, we suspect that the PTRN-1 puncta might not anchored in such a manner; rather, they might be free-floating within the neurite but slow-moving due to crosslinking between bundled MTs and perhaps interactions with other structures within the neurite.

We have therefore altered sentences that had said that PTRN-1 “anchors” MTs to state instead that it “stabilizes” them. From the Introduction, we have also removed the word “immobile” from the following sentence: “Live imaging of the *C. elegans* CAMSAP homolog, PTRN-1, in cell co-labeled with fluorescence-tagged MTs indicates that PTRN-1 localizes to MT associated puncta throughout neuronal processes.” We changed “anchored” to “localized” in describing the implications of the similar localization between PTRN-1::tdTomato and PTRN-1(ΔCKK)::tdTomato in the Discussion.

*4) At several places in the manuscript the authors state that PTRN-1 stabilizes microtubules. Their main argument is that colchicine acts at a lower concentration. This result only says that the loss of this protein makes the microtubules more sensitive to the drug*.

To strengthen our claim that the increased sensitivity of the *ptrn-1* mutant to neurite sprouting during growth on colchicine indicates that PTRN-1 stabilizes MTs in neurites, we have added tissue-specific rescue to these experiments (Figure 5). These data show that PTRN-1 functions in the neurons to protect MTs from the effects of growth on colchicine.

Next, to avoid over-stating the implication of the grow on colchicine experiments, we have made the following alterations to the text that, we hope, clarify the interpretation of these experiments:

The sentence “Again, these synthetic interactions between *ptrn-1* loss-of-function and colchicine indicates that *ptrn-1* promotes MT stabilization in both the 11-pf and the 15-pf neurons” was replaced with “Taken together, these data indicate that loss of *ptrn-1* enhances sensitivity to colchicine cell-autonomously in both the 11-pf and the 15-pf neurons. This enhanced sensitivity to colchicine likely reflects reduced MT stability in the *ptrn-1* mutant, suggestive that PTRN-1 promotes MT stabilization.”

In addition, our deduction that PTRN-1 stabilizes MTs is inferred from the consideration of various other data in addition to the neurite sprouting on colchicine experiments. First, previous studies indicate that *Drosophila* Patronin and the human CAMSAP proteins promote MT stabilization (Meng et al., 2008; Goodwin et al., 2010; Tanaka et al., 2012; Wang et al., 2013). In adjusting the manuscript to answer Major Point 1, we hope we have also more clearly attributed the known function of CAMSAP proteins to these previous works. Second, PTRN-1::tdTomato puncta colocalize with EMTB::GFP foci after acute colchicine treatment whereas PTRN-1(ΔCKK)::tdTomato puncta generally do not. As we state in the text, we believe these data, when considered alongside the excellent characterizations of *Drosophila* Patronin and human CAMSAP proteins, indicate that PTRN-1 itself binds and stabilizes MTs.

*Elsewhere the authors state that they see no difference in structure or number in microtubules except in the PLM. One would expect that the number of microtubules would go down. This needs to be explained*.

We performed detailed characterization of MTs with EM in the PLM neuron but not in other neurons for two reasons. First, PLM has a very well characterized MT number and organization. Previous studies from the Goodman lab and the Chalfie lab have shown that the PLM neuron in wild-type animals has 25-50 MTs per cross-section in young adult animals sectioned between the gonad and the anus (Chalfie & Thomson 1979; Chalfie & Thomson 1982; Cueva et al. 2012, our unpublished data). Second, the unique position and shape of the PLM neurites allow us to unambiguously identify them without reconstructing the entire neuron.

In contrast to PLM, neurons in the ventral nerve cord generally have around 4 MTs per cross section (Chalfie & Thomson, 1979), and we observed as few as one MT in some VNC processes in a wild-type animal. In addition, because there are many neurites in the nerve cord, it is impossible to identify them without a complete reconstruction, which is a huge effort that was only achieved a couple of times in the history. Therefore, due to the large variability of MT numbers between neurites and the inability to identify individual neurons, we did not attempt to compare the number of MTs in the VNC between wild-type and the *ptrn-1* mutant. We did state that MT shape and protofilament number appeared grossly wild-type, but with so few MTs examined, perhaps we should not have attempted to make any statement regarding the VNC MTs from our EM data. Therefore, we have removed Figure 5—figure supplement 1 and the following sentence:

“Electron microscopy of a section of the VNC in the *ptrn-1(tm5597)* mutant did not reveal any gross abnormalities in MT shape or protofilament number (Figure 5—figure supplement 1), but more extensive analysis will be required to determine if there are differences in MT number or length.”

Additionally, in the Abstract, the sentence “Electron microscopy revealed that *ptrn-1* null mutants have a reduced number of MTs and abnormal MT organization” was amended to “Electron microscopy revealed that *ptrn-1* null mutants have a reduced number of MTs and abnormal MT organization in the PLM neuron.”

Of note, the animals prepared for our EM were grown at 25°C to be consistent with all of the other experiments in the manuscript, whereas Chalfie & Thomson, (1979), used animals grown at 20°C. The reduction in MT number at higher temperature is consistent with our previous unpublished observations.

*5) The authors fail to provide statistics for their assertions. The note inadvertently left in the methods shows that at least one of the authors was aware of this problem*.

The Statistics section that summarized the statistical analyses in the Methods was deleted, and the test used in each case was stated in the figure legends. Also, statistical analyses were added to Figure 2 (through the addition of Figure 2—figure supplement 2), Figure 5, Figure 5, and Figure 7.

*6) A large part of the manuscript uses two novel fluorescent markers-labeled patronin and a labeled microtubule-binding domain. It would significantly strengthen the paper to validate these markers. Further, experiments using these markers (colocalization) are descriptive-some quantitative data should be provided*.

We had taken several steps to confirm that the label PTRN-1 is not aberrantly localized. First, we show three different fluorescence labeled PTRN-1 constructs: PTRN-1a::YFP, GFP::PTRN-1a, and PTRN-1a::tdTomato. PTRN-1 tagged at either the N-terminus or the C-terminus and using different fluorophores with different amino acid linkers all exhibited similar localization in multiple neurons of different classes (Figure 1, Figure 1—figure supplement 2, Figure 1—figure supplement 3). This consistency in the localization of labeled PTRN-1 is suggestive that this localization is biologically relevant. Second, we used transgenic lines expressing *ptrn-1a::yfp* to show cell-autonomous rescue for the mislocalized synaptic vesicles and incomplete commissure in the PLM neuron (Figure 6). These data showed that at least some of the labeled PTRN-1 constructs were functional.

To provide additional validation of the labeled PTRN-1, we used the *ptrn-1a::tdTomato* transgene, which is shown in Figure 1, Video 1, Video 2, Figure 1—figure supplement 3, Figure 2, Figure 3, and Figure 3—figure supplement 1, for cell-autonomous rescue of the reduction of EBP-2 movements (Figure 4). We hope that by showing the transgene we used the most for localization data is able to rescue the *ptrn-1* phenotype, we alleviate the concern that the localization is spurious.

The ensconsin microtubule binding domain (EMTB) has been well characterized (22; 11), and it has been used in numerous studies to label microtubules in vivo (34; 59; 62). Our data show that EMTB::GFP localization in the muscle cell is strikingly similar to the localization of the *C. elegans* EMAP protein ELP-1, as described above (Hueston et al., 2009). To further confirm that EMTB::GFP labels MTs in our system, we constructed a strain coexpressing EMTB::GFP and ELP-1::mCherry in the body wall muscle (Figure 8, below)**.** Both fluorescent proteins localize along filaments. We found EMTB::GFP to more consistently label long stretches of the filamentous structures, but the ELP-1::mCherry generally colocalized with the EMTB::GFP, labeling short stretches of the EMTB::GFP filaments. These filaments are MTs. We feel that the request to validate EMTB::GFP was likely elicited by inadequate explanation and citationin the text. We have therefore expanded the introduction of the EMTB::GFP marker in the text as follows:

“To examine whether PTRN-1 binds MTs in neurites, we co-expressed PTRN- 1a::tdTomato with the MT-binding domain of ensconsin fused to GFP (EMTB::GFP)(37; 10; 22). EMTB::GFP, which binds dynamically along the side of MTs, has been previously used to visualize MTs in vivo (11; 34; 59; 62). In *C. elegans* neurons, it generally exhibited continuous fluorescence throughout neuronal processes.”

We also added the reference included below in the Results:

“Since EMTB::GFP normally binds to the sidewalls of MTs (22), this change in its distribution confirms that colchicine treatment led to MT depolymerization, as expected.”Author response image 1.The body wall muscle of an animal oc-expressing EMTB::GFP and ELP-1::mCherry. The two MT-binding proteins colocalize to the MT filaments throughout the muscle cells. Scale bar = 5 μm.

*7) The authors should more clearly define in the abstract and elsewhere which prtn-1 phenotypes are derived from the upregulation of DLK-1 signaling, i.e., which phenotypes are rescued by loss of DLK-1. DLK-1 is not mentioned in abstract or introduction, which gives the impression that the neurite morphology and synaptic vesicle location defects result directly from changes in MT structure caused by loss of PTRN-1*.

To better understand the interaction between *dlk-1* and *ptrn-1,* we examined the affect of *dlk-1* on EBP-2::GFP movements in the PHC dendrite (Figure 7). The following description of these data was added to the Results section:

“Finally, we asked whether the reduction in EBP-2::GFP movements in the *ptrn-1* mutant is also dependent on *dlk-1.* Interestingly, we observed an increase of roughly two-fold in EBP-2::GFP movements in the PHC dendrite in the *dlk-1* single mutant relative to wild- type but had no affect on the orientation (Figure 7**)**. There was a trend for the *dlk-1; ptrn-1* double mutant to have increased EBP-2::GFP movements relative to the *ptrn-1* single mutant, though this difference was not statistically significant. However, the *dlk-1; ptrn-1* double mutant exhibited fewer movements than the *dlk-1* single mutant (Figure 7). This intermediate phenotype in the *dlk-1; ptrn-1* double mutant indicates that the DLK-1 pathway is not the only mechanism required for the *ptrn-1* mutant phenotypes, and it is suggestive that *dlk-1* functions partially in parallel to *ptrn-1* to influence EBP- 2::GFP movements in the PHC neuron.”

To further strengthen the DLK-1 section, we have added Figure 7—figure supplement 1, which shows that *pmk-3,* like *dlk-1*, suppresses two of the defects observed in the *ptrn-1* mutant – the incomplete PLM commissure formation and the loss of synaptic vesicles from the synaptic patch.

We have edited the Abstract as follows:

“Further, *ptrn-1* null mutants exhibited aberrant neurite morphology and synaptic vesicle localization that is partially dependent on *dlk-1*.” Slight changes were made to the wording of the abstract to keep the length to 150 words.

*The authors state that PTRN-1 ‘anchors’ microtubules in neurons and muscle based on the observations that PTRN-1 puncta colocalize with MT puncta following colchicine treatment and that the PTRN-1 CKK deletion mutant often does not colocalize with MTs. As stated, this conclusion is perhaps too strong given that ptrn-1 mutants have a relatively minor, but significant, MT defect. I believe that the conclusion is reasonable, but that it should be stated more conservatively*.

We appreciate the point that we do not present evidence that PTRN-1 itself anchors MTs, and we should not have implied that does. We think the data show that PTRN-1 stabilizes MT foci in both neurites and muscles, since full length PTRN-1 puncta colocalize with EMTB::GFP puncta after acute colchicine exposure whereas PTRN-1(ΔCKK) puncta generally do not. At the sarcolemma, the rows of PTRN-1a::tdTomato and EMTB::GFP puncta can be seen both before and after the acute colchicine exposure, indicating that PTRN-1 localizes to sites of MT anchorage near the muscle membrane. Considering this, along with the fact that the size, spacing, and abundance of PTRN-1 puncta plus EMTB::GFP in the PVD dendrite appears unaffected by acute colchicine exposure, we might have inferred that MTs are likewise anchored at (though not necessarily by) PTRN-1 puncta in the neurites. However, because of the useful suggestion that we take longer movies of PTRN-1::puncta, we now know that many PTRN-1::tdTomato puncta drift in the PVD dendrite, moving at a rate of roughly 1-2 μm per minute. In light of this Major Point and the 40 minute movie of PTRN-1::tdTomato, wherever we described the MTs as being “anchored” in the neurites, we replaced “anchored” with “stabilized.”